# Unravelling the transcriptome of the human tuberculosis lesion and its clinical implications

Kaori L. Fonseca[1,2], Juan José Lozano[3,4], Albert Despuig [1,2], Dominic Habgood-Coote [5], Julia Sidorova[3,4], Diego Aznar [1,6], Lilibeth Arias [1,2], Álvaro Del Río-Álvarez[7,8,9], Juan Carrillo-Reixach[7,8,9], Aaron Goff[10], Leticia Muraro Wildner[10], Shota Gogishvili[11], Keti Nikolaishvili[11], Natalia Shubladze [11], Zaza Avaliani[11,12], Gustavo Tapia[13,14], Paula Rodríguez-Martínez[13,14], Pere-Joan Cardona [1,2,6,15], Federico Martinón-Torres [2,16,17,18], Antonio Salas [2,18,19,20], Alberto Gómez-Carballa [2,18,19,20], Carolina Armengol [7,8,9], Simon J. Waddell [10], Myrsini Kaforou [5,21], Anne O'Garra [22,23], Sergo Vashakidze[11,24,25] & Cristina Vilaplana [1,2,6,15,25] ✉

The tuberculosis (TB) lesion is a complex structure, contributing to the overall spectrum of TB. We characterise, using RNA sequencing, 44 fresh human pulmonary TB lesion samples from 13 TB individuals (drug-sensitive and multidrug-resistant TB) undergoing therapeutic surgery. We confirm clear separation between the TB lesion and adjacent non-lesional tissue, with the lesion samples consistently displaying increased inflammatory profile despite heterogeneity. Using weighted correlation network analysis, we identify 17 transcriptional modules associated with TB lesion and demonstrate a gradient of immune-related transcript abundance according to spatial organization of the lesion. Furthermore, we associate the modular transcriptional signature of the TB lesion with clinical surrogates of treatment efficacy and TB severity. We show that patients with worse disease present an overabundance of immune/inflammation-related modules and downregulated tissue repair and metabolism modules. Our findings provide evidence of a relationship between clinical parameters, treatment response and immune signatures at the infection site.

Tuberculosis (TB) is an infectious disease caused by *Mycobacterium tuberculosis* (*Mtb*), and a major cause of ill-health and mortality worldwide. Globally, TB chemotherapy is successful in 85% of drug-sensitive (DS) TB cases[1]. Nevertheless, there is a fraction of patients who will fail treatment and are therefore prone to disease relapse and death, especially in multi drug-resistant (MDR) TB cases[1]. The formation of granulomas is a hallmark of TB and is crucial for containing and controlling the spread of *Mtb* within the host[2], involving numerous immune cell types[3]. The existing literature has demonstrated a high degree of heterogeneity in TB granulomatous lesions[3]. Animal studies involving macaques have provided valuable information on granuloma nature and evolution, showing high diversity even within the same host with different grades of bacteria clearance[4]. Moreover, this diversity is observed over the course of infection[5]. Preclinical studies are key to understanding how TB lesions evolve, as human studies cannot provide this information unless using surrogates, such as 18-F-fluorodeoxyglucose positron emission tomography-computed tomography (18F-FDG-PET-CT), as demonstrated by Malherbe et al. when

**Fig. 1 | Overall study plan.** Overview of the analysis undertaken in the study. Figures associated with each objective are stated.

correlating images of individuals with TB obtained using this method with bacillary load[6,7]. Additionally, mycobacterial culture from resected granuloma tissue demonstrated that a subset of individuals still harboured live *Mtb* bacilli despite preoperative microbiological clearance in sputum, in both DS- and MDR-TB[8,9].

The development of lung cavitary lesions from granulomas is a key aspect of the TB pathogenesis, associated with increased transmission rates and poor outcomes[10]. Human lung biopsies from TB lesion are limited[11] and the host factors that drive cavitary lesion formation or indicate poor clinical outcomes remain unknown. The resection of human pulmonary lesions during therapeutic surgeries or autopsies has provided insights into TB lesion architecture, and local immunopathology which may contribute to the emergence of MDR *Mtb* populations[8,12].

Several studies on human TB granuloma tissue imaging and the computerized quantification of cells and molecules at RNA and protein levels have provided valuable insights into granuloma structure, cellular composition, and immune responses[13–16]. Recently, Carow and colleagues compared human pulmonary TB lesions to those in patients with sarcoidosis, revealing significant differences in immune cell distribution and, consequently, in their immunological microenvironments[17]. Subbian et al demonstrated a molecular correlation of immune responses to the heterogeneity of granuloma samples from four MDR-TB cases, diversity that the authors linked to lesion maturation[18]. Marakalala et al. suggested that the response to *Mtb* might be shaped by the anatomical localization within the granuloma[19]. Dheda et al. were the first authors to characterize the transcriptional response at anatomically different locations within the granulomas of 14 MDR-TB cases[11]. They showed the

cavity wall as the main source of pro-inflammatory activity compared to the lesion centre. Finally, a recent study constructed a spatial cell atlas using 6 patients' samples (two DS-TB and one MDR-TB patient undergoing surgery, and three autopsies) to map granuloma structure and composition and contrast it with the peripheral immune responses[20].

In this study we characterize the cellular of human TB pulmonary lesions from DS-TB and MDR-TB patients who underwent surgery and show their link to clinical and microbiological surrogates of TB severity and treatment response (Fig. 1).

## Results

### The human TB lesion signature shows a distinct and heterogeneous transcriptional profile as compared with non-lesional lung tissue

In our study, outlined in Fig. 1, we collected 48 samples from 14 individuals and analysed 44 paired samples from 13 individuals (6 DS-TB and 7 MDR/XDR-TB) to evaluate the human TB lung granuloma transcriptomic changes by RNA sequencing.

Although the patients included in this study exhibited normal to high BMI, low CRP levels, relatively low SGRQ scores and were considered microbiologically cured, they nonetheless required lung resection surgery due to the persistence of TB cavities.

We analysed total RNA from three different sections: Central Lesion (C; $n = 6$), Internal Wall (I; $n = 12$) and External Wall (E; $n = 13$) collected from each patient's lesion biopsy. Fewer C- samples could be analysed compared to I and E, due to poorer RNA recovery. Additionally, surrounding non-lesional (NL) tissue from the involved

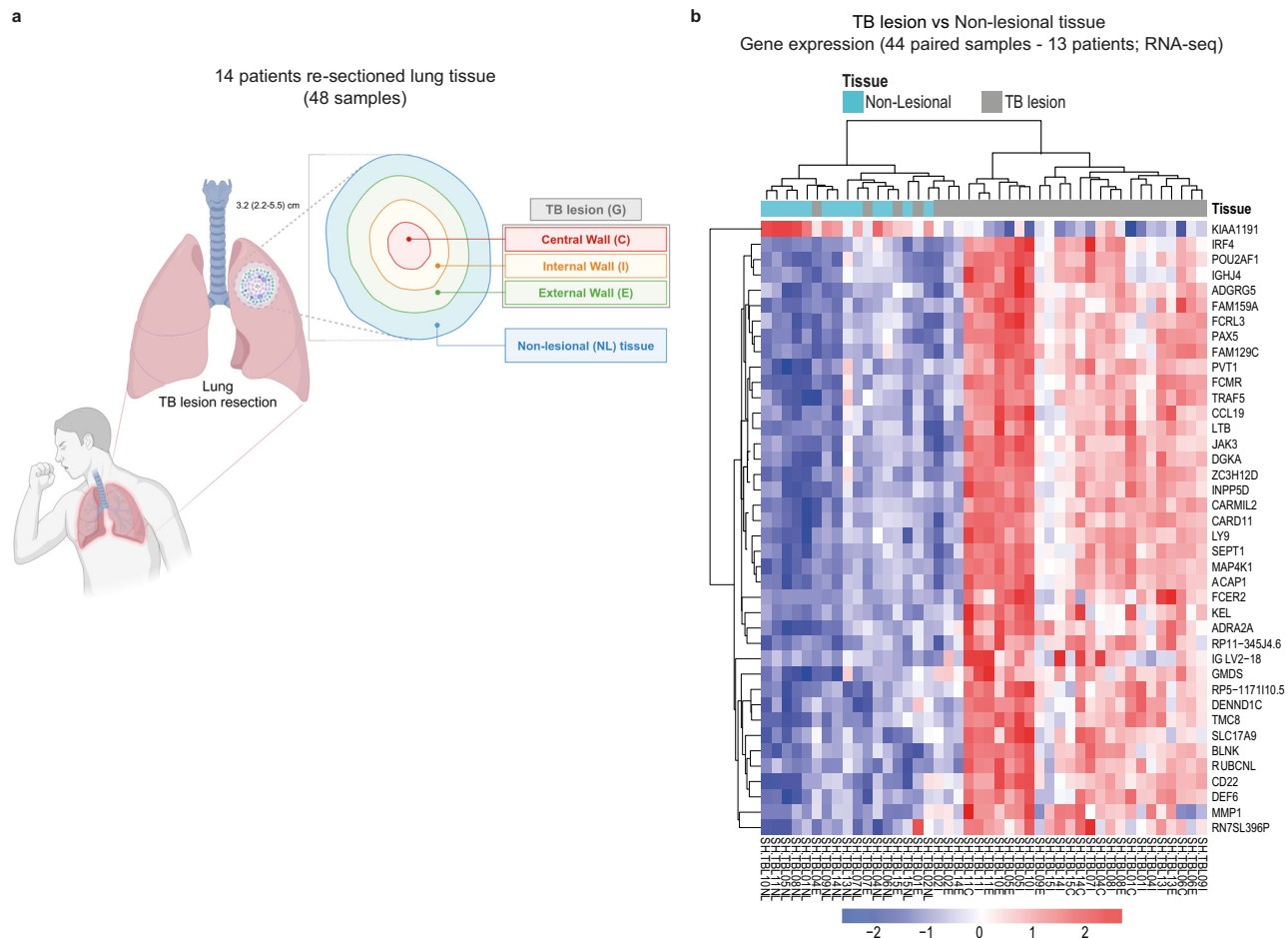

**Fig. 2 | The human TB lesion signature shows a distinct and heterogeneous transcriptional profile as compared with non-lesional lung tissue.** TB lesion samples were collected from each patient included in the SH-TBL cohort: central lesion (C), internal wall (I) and external wall (E) and, altogether, samples from each patient represent the human TB lesion. An additional sample from surrounding non-lesional lung tissue (NL) was also collected from the same patient as control (**a**). 48 samples from 14 patients (6 DS-TB and 8 MDR/XDR-TB) were RNA sequenced to evaluate the human TB lung lesion transcriptomic changes. A set of 4630 DEGs was identified after comparing the human TB lesion counts with NL lung

tissue expression, using DESeq2 with adjusted $p < 0.05$. **b** heatmap depicts the top 40 DEGs ranked by the adjusted $p$-value comparing the human TB lesion versus NL lung tissue expression profiles (44 paired samples from 13 patients). The intensity of each colour denotes the standardized ratio between each value and the average expression of each gene across all samples. Red pixels correspond to an increased abundance of mRNA in the indicated sample, whereas blue pixels indicate decreased mRNA levels. Source data are provided as a Source Data file. Image in (**a**) was created in BioRender. Vilaplana, C. (2025) https://BioRender.com/x16o926.

lung was collected as a comparator ($n = 13$) (Fig. 2a). Patients were matched according to their sex and *Mtb* drug-sensitivity classification to avoid potential confounding factors (Supplementary Table 1). Moreover, clinical and demographic data, and resected TB lesion characteristics and pathology were assessed at the time of surgery and are reported for each participant (Supplementary Table 1 and Supplementary Data 2).

We found a total of 4630 significantly differentially expressed genes (DEGs), using DESeq2 with adjusted $p \leq 0.05$ (Supplementary Fig. 1a). Of these, 2496 genes were over-expressed in lesion tissues, whereas 2134 were under-expressed, as compared to NL lung tissue (Supplementary Fig. 1a). The top 40 ranked DEGs clearly separated lesion samples from NL lung samples (Fig. 2b), showing distinct transcriptional profiles for the two tissues. Among them, genes involved in immune system/cytokine signalling (*IRF4, CCL19, LTB, JAK3, INPP5D, FCER2, MMP1*) and B cell activation and differentiation (*CD22, BLNK, CARD11*) were over-expressed, suggesting an inflammatory signature in the TB lesion.

Seven TB lesion samples clustered together with NL tissue samples, consisting of six samples from the external compartment and one

from the internal compartment. This observation may suggest a transitional transcription profile across the lesions, particularly evident in the external tissue due to its proximity to the NL samples, but also not discarding the heterogeneity in the transcriptional profiles of the lesions (Fig. 2b).

Altogether, our data show a distinct segregation of the TB lesion when compared to the NL lung tissue with respect to an inflammatory profile, as previously proposed[11]. Our findings also indicated a range of molecular diversity within the TB lesion samples, prompting our decision to delve deeper into the heterogeneity at a transcriptional level.

### Compartments within the TB lesion reveal distinct gene expression profiles with an enriched inflammatory response across the lesion

To further explore the TB lesion heterogeneity and investigate the contribution of each compartment, we first performed an enrichment analysis derived from single sample Gene Set Enrichment analysis (ssGSEA) using the top 40 DEGs discriminating the TB lesion from the NL lung tissue. The expression of these genes in the different tissue compartments revealed a more pronounced enrichment score of

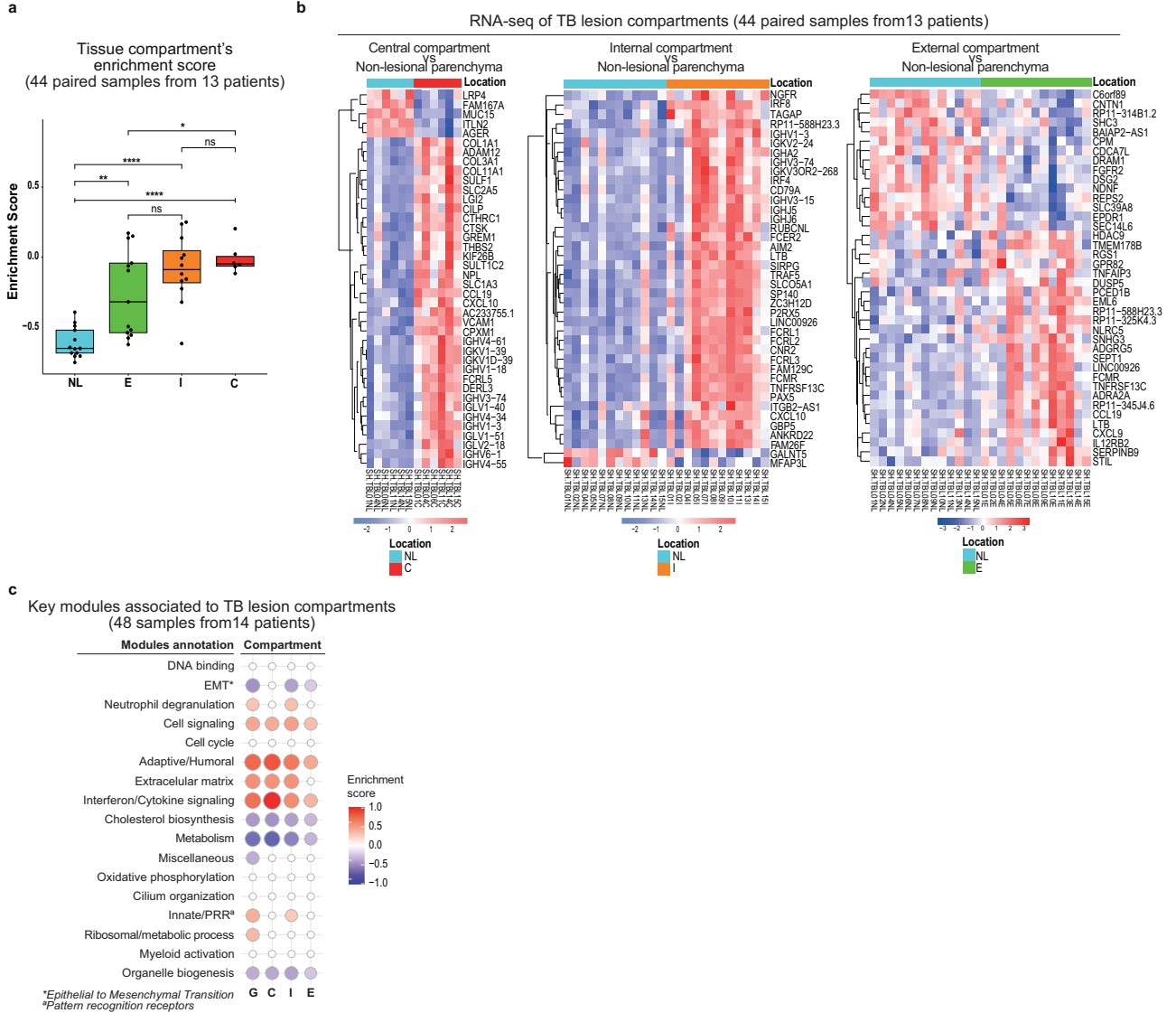

**Fig. 3 | The human TB lung lesion compartments have different gene expression profiles and are enriched for immune inflammatory response pathways.** **a** show the enrichment score derived from single sample analysis GSEA using the top 40 genes discriminating TB lesion (G) from NL lung tissue. Data on the enrichment for each compartment (C, I, E and NL) are represented as medians with an interquartile range (IQR). Boxplots show minimum and maximum values, the interquartile range (IQR, 25th to 75th percentile), and the whiskers representing 1.5 times the interquartile range. Outliers are indicated as individual points outside the whiskers. Statistical analysis was performed by applying the two-sided *t*-test. Statistical differences refer to a *p*-value < 0.05. In (**b**) the heatmaps show differences in the top 40 ranked genes from DESeq2 with adjusted *p* < 0.05 by separately comparing the central (C), internal (I) and external (E) compartments with the NL lung tissue gene expression derived (). The intensity of each colour denotes the standardized ratio between each value and the average expression of each gene across all samples. Red pixels correspond to an increased abundance of mRNA in the indicated sample, whereas blue pixels indicate decreased mRNA levels. **c** pictures modular transcriptional of the seventeen modules of co-expressed genes derived from WGCNA for our TB lesion dataset separated by compartment. Fold enrichment scores derived using QuSAGE are depicted, with red and blue indicating modules over or under expressed compared to the control. Only modules with fold enrichment (FDR) < 0.1 were considered significant. Source data are provided as a Source Data file.

these DEGs in central and internal lesion samples, suggesting that these two compartments might be the main contributors for the overall TB lesion transcriptional signature (Fig. 3a).

Next, we compared the expression profiles derived from each TB lesion compartment with the NL tissue. The list of DEGs (DESeq2 with adjusted *p* ≤ 0.05) for the C, I and E *vs* NL tissue comparisons respectively constituted 3228 (1539 genes were over-expressed, whereas 1689 were under-expressed); 5275 (2676 over-expressed and 2599 underexpressed); and 1045 genes (552 over-expressed and 493 underexpressed) (Supplementary Fig. 1b). For central and internal compartments, the hierarchical clustering of the 40 most significant DEGs showed an evident separation when compared each compartment against the NL lung tissue (Fig. 3b). Though less noticeable, the

external compartment was still distinguishable from the NL tissue. Therefore, the magnitude of differential expression relative to NL decreased gradually towards the edge of the TB lesion structure, including between adjacent compartments (Fig. 3b and Supplementary Fig. 2).

Among the highly variable genes in central lesion, we found genes involved in the immune system/cytokine signalling (*CCL19, CXCL10*) to be upregulated in comparison to the NL tissue (Fig. 3b). On the other hand, we found extracellular matrix organization-related genes to be downregulated (*LRP4, MUC15*), while others were upregulated (*ADAM12, CTSK*). Moreover, collagen-encoding genes (*COL1A1, COL3A1, COL11A1*) were upregulated in the central compartment, which could reflect the fibrosis observed in all patients' lesions

(Supplementary Data 2); as well as of genes associated to immunoglobulin heavy and light chains (*IGHV4 − 61, IGKV1 − 39, IGKV1D − 39, IGHV1 − 18, IGHV3 − 74, IGLV1 − 40, IGHV4 − 34, IGHV1 − 3, IGLV1 − 51, IGLV2 − 18, IGHV6 − 1, IGHV4 − 55*), related to humoral immunity (Fig. 3b). Furthermore, genes involved in complement fixing (*C1QA, C1QB, C1QC*) were significantly upregulated, although not among the top 40 DEGs (Supplementary Data 1). For the internal compartment, genes involved in the immune system/cytokine signalling (*LTB, FCMR, AIM2, CXCL10, IRF8, IRF4*) were upregulated compared to NL (Fig. 3b). Furthermore, immune system/cytokine signalling genes (*LTB, CCL19, CXCL9, TNFAIP3, TNFRSF13C, FCMR, AIM2, CXCL10*) were overexpressed in the external compartment relative to NL (Fig. 3b), evidencing an inflammatory signature throughout the lesion.

We then applied weighted gene co-expression analysis (WGCNA) to perform a modular analysis of co-expressed genes in the TB lesions and in the three compartments separately, comparing all samples to NL control tissues. We identified 17 modules from co-expression networks related to the whole human TB lesion (Fig. 3c and Supplementary Data 2). The identified TB lesion modular signature showed that neutrophil degranulation, cell signalling, adaptive/humoral immunity, extracellular matrix, interferon/cytokine signalling, and innate/pathogen recognition receptors (PRR) modules were overabundant. These observations were consistent throughout the compartments, except for the neutrophil degranulation and innate/PRR modules, which were apparent in the total lesion and the internal lesion only, but not in the central or external lesions (Fig. 3c). Conversely, the Epithelial to Mesenchymal Transition (EMT), cholesterol biosynthesis and metabolism modules were found to be underabundant in the whole lesion and external and internal, but not in the central compartment. In addition to cholesterol and metabolism, the organelle biogenesis module was underrepresented, across all compartments, suggesting that some pathways present in the healthy lung are diminished in the lesion (Fig. 3c).

To broaden our understanding on the distribution of the immune response among the modular signature in the whole lesion and its compartments, we have used the LM22 signature matrix to profile the distinct human hematopoietic cell populations[21]. We found that the adaptive/humoral and the innate/PRR modules presented most of the genes related to the immune populations (Supplementary Fig. 3a), with said populations also varying in proportions across TB lesion compartments (Supplementary Fig. 3b). By expanding these modules, we found that most of the submodules composing them were significantly differently enriched when comparing the compartments, particularly for the adaptive/humoral submodules (Supplementary Fig. 4a).

In summary, our results showed a significant enrichment of modules related to inflammation, including pathways of innate immunity in the TB lesion, the central and internal compartments, and of adaptive/humoral immunity across all compartments. Meanwhile a decrease in modules related to extracellular matrix organisation and cholesterol biosynthesis and metabolism was observed in the lesion. Furthermore, profiling of the LM22 populations as well as the expansion of the adaptive/humoral and innate/PRR modules revealed a differential distribution of the immune cells in the different compartments, contributing to the identified modular signature.

### Patients' clinical status is associated with differential modular transcriptomic profiles in TB lesions

The heterogeneity in the host immune response to infection, considering the involvement and contribution of physically distinct compartments, together with the bacteria and the inflammatory environment, defines granuloma fate and disease manifestation[19,22]. Hence, we next aimed to associate the modular signature changes in the TB lesion (considering the three compartments together) with clinical data (Supplementary Data 2), using surrogates of treatment response and disease severity (SGRQ symptoms sub-score; being a fast

or slow sputum culture converter; DS vs MDR-TB case; being a relapse or new TB case and number of lesions present in the CXR). We quantitatively tested the association of each clinical parameter with each of the significant module's eigengene (ME) expression patterns (Wilcoxon $p \leq 0.05$).

Regarding the sputum culture conversion (SCC), the modular signature of the TB lesion revealed a significant association of DNA binding and interferon/cytokine signalling modules with SCC, with the enrichment of these modules being significantly higher in those individuals converting the sputum culture later (FDR < 0.1; Fig. 4a, b). No significant modular expression was found to be associated with *Mtb* drug sensitivity of the individuals, relapsed or new cases, or number of lesions (Supplementary Data 2).

When considering the severity of TB disease, in terms of a higher presence and severity of symptoms, we found that DNA binding, neutrophil degranulation, interferon/cytokine signalling, cholesterol biosynthesis and myeloid activation modules were significantly overabundant and associated with higher SGRQ symptoms score (FDR < 0.1; Fig. 4a and c), pointing to higher inflammation status with more severe disease manifestation. In contrast, the EMT module was significantly underabundant in these individuals' TB lesions (Fig. 4c). When stratifying against the clinical data, results showed that there was no clustering of the clinical surrogates with neither the central nor the internal compartments (Supplementary Fig. 5).

Further examination of the associations between submodules and severity correlates revealed that only three submodules exhibited statistically significant differences among clinical surrogates (Supplementary Fig. 4b), derived from the innate/PRR module: the submodule Response to inflammation, linked to neutrophils and granulocytes, with genes related specifically to response to IL-1 and type II IFN and neutrophil chemotaxis; and the Innate response regulation submodule, linked to neutrophil, monocytes and macrophages, with genes related specifically to immune response activation and regulation. We also observed an enrichment in the CD4 + T helper lymphocyte response submodule. derived from the Adaptive/humoral module, linked to antigen-presenting cells and CD4 T cells, with genes specifically related to the regulation of T cell activation, lymphocyte differentiation and regulation of the adaptive immune response.

To gain further insights into the differences found between clinical surrogates, we next identified a set of seven transcription factors differentially expressed (*ETV7, STAT1, AR, SOX5, ERG, ASCL2* and *PRDM5*) between fast and slow SCC and patients with less severe or more severe symptoms. Transcription factors corresponding to the IFN/cytokine signalling module were overexpressed in slow converters and/or more severe patients, whereas transcription factors belonging to the EMT module had higher expression in patients with less severe symptoms, complementing the modular analysis (Supplementary Fig. 6a, b and Supplementary Table 2).

To support RNA sequencing data, we validated by immunohistochemistry the protein products of three genes significantly expressed between TB lesion and NL tissue. In our data set, CXCL9, GBP5 and STAT1 are representative genes from the module with the highest enrichment in the whole TB lesion associated with TB surrogates of severity. We quantified the presence of the respective proteins in TB patient lesions compared to non-TB controls and found a significantly higher expression of these proteins in the TB patient lesions (Fig. 5a−d).

All these data reinforce our findings from the transcriptional comparison between TB lesion and NL tissue, showing that a slower SCC, reflecting a poorer response to treatment and slower clearance of *Mtb*, and severer TB cases are associated to an increased inflammatory response at site.

## Discussion

The immune response to *Mtb* constitutes a complex and dynamic interaction between the host immune system, the bacteria and lung

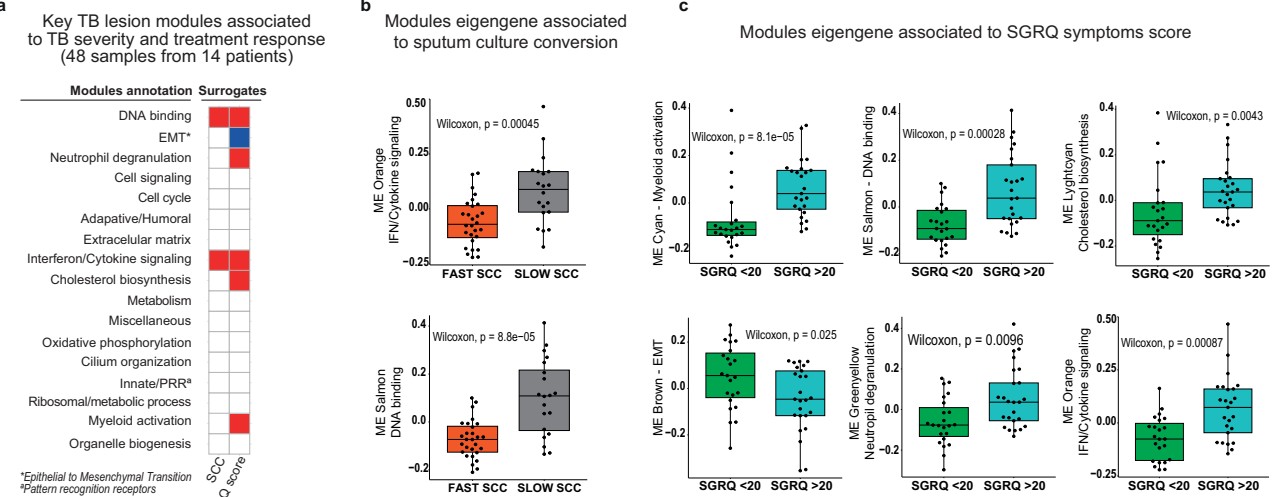

**Fig. 4 | TB lesion modular transcriptional signature correlates with TB clinical and microbiological characteristics revealing differential responses between patient's group.** Modular analysis of RNA-seq data from TB lesions of 14 patients. Patients were clinically defined accordingly to sputum culture conversion (SCC) and TB disease impact on lung function, measured using the Saint George's Respiratory Questionnaire (SGRQ) symptom score, as surrogates of treatment response and TB severity. Heatmap represent the key TB lesion modules significantly associated to individual's clinical surrogates of TB severity and treatment response (**a**). Fold enrichments were calculated for each WGCNA module using hypergeometric distribution to assess whether the number of genes associated with each clinical status is larger than expected. Fold enrichment scores derived using QuSAGE are depicted, with red and blue indicating modules over or under expressed compared to the control. The colour intensity represents degree of perturbation. Modules with fold enrichment scored FDR *p*-value < 0.1 are considered significant. **b**, **c** show TB individuals' stratification according to SCC (fast *n* = 28 or slow converters *n* = 20) and SGRQ symptom score (low impact if SGRQ < 20 with *n* = 23 or high impact if SGRQ > 20 with *n* = 25), respectively, and the significant association using their corresponding derived WGCNA significant eigengene modules (ME) (*p* < 0.05). Data are represented as median with an interquartile range (IQR). Boxplots show minimum and maximum values, the interquartile range (IQR, 25th to 75th percentile), and the whiskers representing 1.5 times the interquartile range. Outliers are indicated as individual points outside the whiskers. Statistical analysis was performed by applying the two-sided Wilcoxon-rank sum test. Source data are provided as a Source Data file.

microenvironment. Throughout infection, the inflammatory response leads to granuloma formation primarily for bacterial containment, while causing extensive tissue remodelling and destruction, which contributes to the clinical spectrum of TB[20]. In our study, we transcriptionally characterised the host response in human pulmonary TB lesions from patients undergoing therapeutic surgery. Fresh human TB lesion specimens obtained from these lung resections (without formalin fixation or paraffin embedding) were transcriptionally profiled using RNA-Seq. Our study provides various advances over previous approaches[18,20,23]. Firstly, we have used a more robust data set with increased individuals numbers, which included 44 TB lesion samples from 13 DS- and MDR-TB patients from the SH-TBL cohort. Secondly, we confirmed a distinct demarcation between the TB lesion and adjacent non-lesional tissue from the same patient lung. Thirdly, we have identified a transcriptional modular signature within TB lesions and linked our findings to clinical/microbiological parameters used as surrogates of TB severity and response to treatment. In individuals with more severe disease, our results showed an increased eigengene expression of pro-inflammatory-related modules and a decreased eigengene expression of tissue organization modules. Strikingly, those individuals with a delayed response to treatment showed an increased DNA binding and interferon/cytokine modular response.

In our view, establishing a link between a slower culture conversion rate (persisting beyond 8 weeks post-treatment initiation) and the presence of more inflamed lesions at treatment's end opens up the potential for refining TB treatment during clinical management.

Granuloma heterogeneity in TB is a well-accepted concept and has been reported in non-human primate models of infection and human lesions[13,14,19,24]. Besides the heterogeneity among samples, we were able to identify a clear pattern across all TB patients compared to their own NL control lung tissue. Among the top 40 DEGs, we found genes predominantly encoding for proteins involved in the

inflammatory processes that orchestrate the antimycobacterial response, as previously reported[25–27]. This included genes as *CCL19*, which expression was found to be increased in mouse lungs post-*Mtb*-infection to induce lymphoid structures[25]; *FCMR*, considered a target for host-directed therapies[26], and the transcription factor *IRF4*, previously found to be required for the generation of Th1 and Th17 subsets of helper T cells and follicular helper T-like cellular responses[27]. Additionally, the overexpression of immunoglobulin genes in the TB lesion suggests their involvement in complement fixation processes, since *C1QA, C1QB* and *C1QC* transcripts were also found to be upregulated in our TB lesion samples. Previously published blood signatures found upregulated levels of *C1QC*[28,29], when comparing baseline to end-of-treatment samples. Moreover, the expression of this gene has been proposed as a disease severity biomarker[30] and linked to poor treatment response[31].

The pro-inflammatory transcriptional signature observed in the TB lesion was eminently due to its central and internal compartments, in line with Dheda's and Marakalala's studies[11,19] Dheda et al. described the pathways involved in different parts of cavitary lesions from 14 failed MDR-TB participants that underwent surgery, pointing to the cavity wall as the main source of pro-inflammatory activity[11]. In line with our findings, they showed that pro-inflammatory pathways were especially over-represented in the cavity wall, including nitric oxide production, reactive oxygen species, IL-1, IL-6, IFN-γ and NF-κβ transcriptional signatures. Furthermore, Marakalala et al. demonstrated a pro-inflammatory centre and an anti-inflammatory surrounding tissue by mass spectrometry and lipid quantification. These authors worked with different types of granulomata from six MDR-TB patients and highlighted the heterogenicity of the lesions[19]. Additionally, Subbian et al. demonstrated using four granuloma samples, the involvement of immune cell signalling and activation, interferon response and tissue remodelling processes in the complex TB granuloma

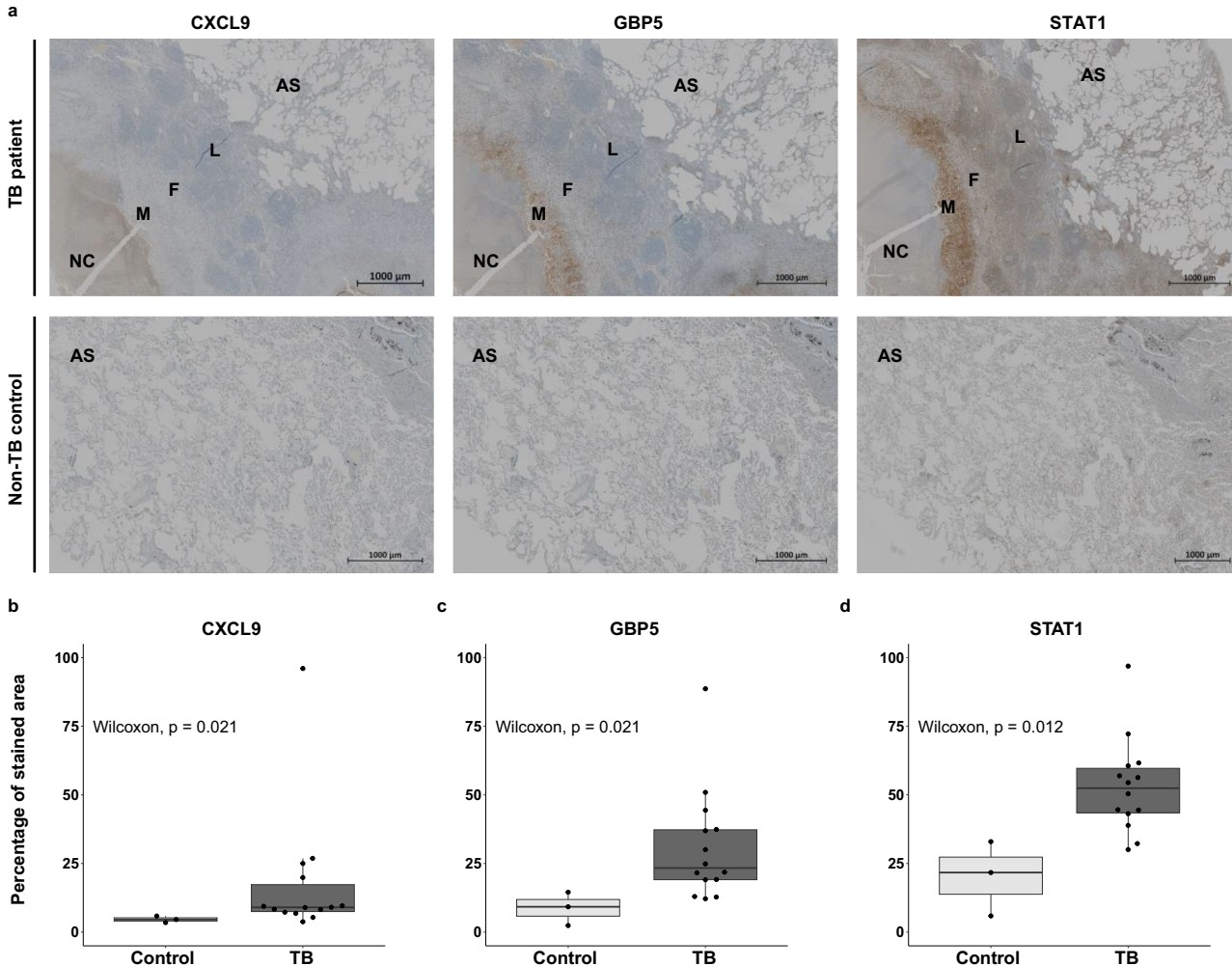

**Fig. 5 | Immunohistochemistry staining of representative genes associated with TB severity reveals higher protein expression in TB compared to non-TB controls. a** shows representative immunohistochemistry staining for CXCL9, GBP5 and STAT1 from the TB lesion of a representative patient (TB-05) compared to a patient presenting bullous emphysema (TB-42), as non-TB control. The top row corresponds to whole sections of the TB lesion (at the left of the images) and of non-lesional tissue (at the right of the images). Scale bars correspond to 1000 μm. NC necrotic core, M macrophage region, F fibrotic region, L lymphocyte-enriched region, AS alveolar space. **b–d** show the quantification of CXCL9, GBP5 and STAT1 protein levels respectively in lesion sections of all TB patient (*n* = 14) compared to the non-TB control tissue sections (*n* = 3). *n* refers to biologically independent tissue sections from different individuals. Data on the percentage of stained area are represented as median with an interquartile range (IQR) Boxplots show minimum and maximum values, the interquartile range (IQR, 25th to 75th percentile), and the whiskers representing 1.5 times the interquartile range. Outliers are indicated as individual points outside the whiskers. Statistical analysis was performed by applying the two-sided Wilcoxon-rank sum test. Statistical differences refer to a *p*-value < 0.05. Source data are provided as a Source Data file.

microenvironment[18]. The TB lesion modular signature that we describe herein, provides comparable and additional data, albeit in aa independent and representative patient cohort and including DS-TB.

We identified 17 modules from co-expressed networks and mapped a TB lesion modular signature, consisting of increased neutrophil degranulation, cell signalling, adaptive/humoral immunity, extracellular matrix, interferon/cytokine signalling and innate/PRR. In our cohort, patients presented advanced TB disease with cavitary TB. We found the neutrophil degranulation module increased in whole TB lesion but not in the central or external lesions, possibly explained by more necrosis in this region, coupled with relatively low RNA abundance in neutrophils. Moreover, the *MMP1* gene was overexpressed in the TB lesion as compared to NL tissue, which might suggest the involvement of neutrophils through the activity of matrix metalloproteinases. In previous studies, MMP-1 was found to be increased in the respiratory secretions from TB patients and to drive extracellular matrix remodelling in a TB murine model[32], and to be differentially expressed in human TB lymph node biopsies compared to control samples in a study by Reichmann et al.[33]. In humans, neutrophil

accumulation in the lungs of individuals with TB and has been correlated with increased lung pathology and consequent disease progression[34,35]. The role of neutrophils in TB disease progression and pathology has been well documented in experimental mouse models[34–37]. Additionally, the extracellular matrix, interferon/cytokine signalling and innate/PRR modules have been reported to be upregulated in blood from individuals with TB[38,39].

Interestingly, we found that the adaptive/humoral module was increased in whole TB lesion samples, corroborating the expression of immunoglobulin heavy and light chains transcripts in both central and internal compartments, as well as the higher proportion of effector B cells across the TB lesion. The enrichment of the adaptive/humoral module, along with increased lymphocytes—particularly effector B cells —in TB lesions, suggests an elevated antibody response. This is consistent with recent findings by Krause et al., who reported abundant B cells and high levels of *Mtb*-reactive antibodies in these lesions[40]. By expanding the adaptive/humoral and innate/PRR modules, we identified genes associated with B and T lymphocyte responses, and innate response regulation. Furthermore, we showed that these responses and

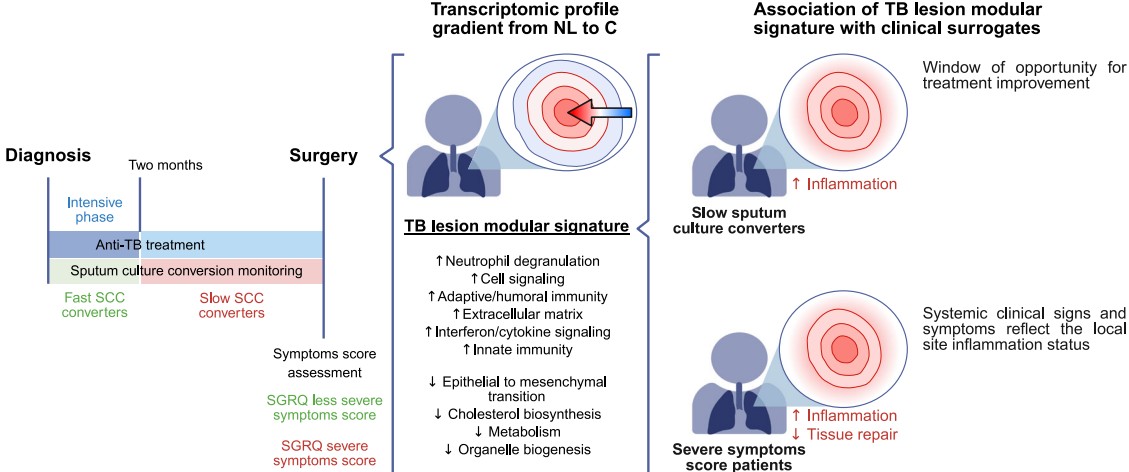

**Fig. 6 | Overview of the main conclusions of this study.** This figure summarizes the key findings of our study. We obtained a modular transcriptomic signature for the TB lesion that follows a gradual increase of differential expression relative to non-lesional tissue towards the center of the lesion, including the enrichment of gene modules associated with the immune response and inflammation within the lesion. The association of a worsened local status of the lesion with clinical and microbiological surrogates contributes to the immunopathological understanding of the disease and may aid in the clinical management of the disease by opening a window of opportunity for the adjustment of treatment. This figure was created in BioRender. Vilaplana, C. (2025) https://BioRender.com/f17r368.

the proportions of their related immune cells are distributed in a gradual manner across various compartments, which appears to play a central role in the modular signature of TB lesions here unveiled. Overall, the enrichment of immune-related modules mostly in the central and internal compartments and the observed cell distributions suggest that, although the central compartment is predominantly necrotic, it may still harbour a lymphocyte component from adjacent tissue. However, due to the lower overall cellularity in this central compartment, the relative abundance of these populations appears higher, reflecting a greater proportion of T cell-associated signals rather than an increased absolute number of T cells. The relative proportions of each cell population across compartments align with previous studies[16,20,41], as the outer portion of the lesion exhibits a higher abundance of lymphocyte populations, whereas macrophage populations are more prevalent in the inner compartments.

Conversely, the EMT, cholesterol biosynthesis and metabolism modules were decreased in TB lesion relative to NL parenchyma. The EMT is linked to wound healing but also to fibrogenesis and scarring[42]. Furthermore, several transcription factors associated with this module were differentially expressed based on disease severity. These include SOX5, involved in chondrocyte differentiation[43]; PRDM5, which has a role in proper extracellular matrix development[44]; ERG, essential in normal hematopoietic stem cell function[45]; and AR, which is involved in the reduction of pro-inflammatory responses in monocyte and macrophages and their M2 polarization[46]. The downregulation of this module in TB lesion along with its transcription factors, and its decreased enrichment in more severe TB cases compared to those with milder symptoms, might suggest a disruption in critical processes needed for tissue repair. We also observed increased cholesterol synthesis in individuals experiencing more severe disease in terms of presenting more pronounced symptomatology. Kim et al. proposed dysregulation of host lipid metabolism caused by *Mtb*, tracing the progression of TB granulomas to caseation, cavitation, and eventual disease transmission[23]. The authors suggested that bacterial components could trigger the host's innate immune system, potentially augmenting the synthesis or storage of host lipids. Consequently, in line with these findings[23] the upsurge in cholesterol synthesis which we observe herein might mirror the impact of *Mtb* on the host lipid metabolism.

The major outcome of our work is the use of unbiased modular analysis to link the transcriptional signature generated from TB lesions with patients' clinical surrogates of TB severity and the time taken to clear *Mtb* in sputum, and thus response to treatment, as summarized in Fig. 6. Our results show an important inflammation component in lesions from individuals presenting with greater severity of disease and slower response to treatment. Inflammation has been described as key for tissue damage and linked to a blood transcriptional signature in individuals suffering from active TB disease even months before being diagnosed[47], and radiographic lung disease extension[5,34,38], decreasing upon treatment[38,48]. Tabone et al. revealed differential responses in the blood transcriptional signature among various clinical TB subgroups following treatment, observing a reduction in the inflammation and IFN modules alongside B and T cell modular signatures accompanying successful treatment[39]. In our study, we found an overabundance of the IFN/cytokine signalling and DNA binding modules associated with severe disease, characterized by worsened symptoms and slower bacterial clearance. The overabundance of the IFN/cytokine signalling module is accompanied by the differential expression of the STAT1 and ETV7 transcription factors in both patients with worse symptoms and slower response to treatment and ASCL2 in patients with worse symptoms. STAT1 plays a main role in TB pathogenesis via the activation of IFN responses[38,49]. On the other hand, ETV7 has been identified as a regulator of inflammatory responses by repressing IFN-induced genes[50,51]. Meanwhile, ASCL2, which initiates the development of T follicular helper cells, has been shown to suppress IFN-γ[52]. This simultaneous differential expression emphasizes the increased inflammatory component characteristic of a worsened disease state, along with the counter-response that attempts to limit the side effects of this devastating inflammation. Additionally, immunohistochemistry showed that the representative genes *CXCL9*, *GBP5* and *STAT1* from the IFN/cytokine signalling module had significantly higher protein levels in TB patient lesions compared to the non-TB control samples, supporting our RNA-seq findings.

Our observations also hint at a potential connection between heightened neutrophil degranulation in severe cases and the damaging mechanisms associated with neutrophil-mediated inflammation[53], suggesting a plausible role for this process in exacerbating the severity of the condition. As our study samples were collected after the end of treatment, all cases examined here could be considered difficult or inadequate responders to treatment. This inadequate response may result in a sustained pro-inflammatory profile at the lesion site or be the consequence of it, and our findings may thus help in future management of disease treatment.

The results suggest that the clinical picture mirrors the inflammation happening at the site of infection and confirms what has been previously seen by others indirectly, both in humans and in experimental animal models. Malherbe et al., showed through 18F-FDG-PET-CT lung scans that some patients still have an increased FDG uptake in the lesions when compared to surrounding healthy tissue after six months of treatment[6], and more recently, the authors have related both a larger burden of disease and a slower rate of reduction in scan metrics with delayed sputum coversion[54]. Our data showed that slow sputum converters present different modular expression profiles in TB lesions when compared to fast sputum converters. To date, the SCC constitutes the only tool endorsed by the WHO to monitor treatment response[55] and can be considered a surrogate of disease severity. Therefore, achieving SCC after two months of starting treatment has been associated with TB cavity persistence[9] and poor prognosis[56,57], and has been proposed and used as a surrogate marker for TB outcomes. Now, thanks to our study, we demonstrate that a sustained pro-inflammatory profile at the lesion site is linked to delayed culture conversion (persisting beyond 8 weeks post-treatment initiation) focusing on a cohort of patients with severe immunopathological disease who underwent therapeutic surgery. This finding implies that delayed culture conversion may serve as a proxy for heightened lesion inflammation and, by extension, worse clinical outcomes. It may thus facilitate the identification of patients who could benefit from enhanced therapeutic strategies, including the incorporation of anti-inflammatory host-directed therapies into standard treatment regimens. Interestingly, measures of microbiological treatment success and clinical severity of disease have also been associated with *Mtb* transcriptional profiles in patient sputa[58], suggesting that the lesion immunopathology described here also impact *Mtb* lung phenotypes.

Our study has some limitations. These TB individuals presented advanced TB disease rendering them candidates for lung resection surgery despite being microbiologically cured. Notably, the surgery was performed not because of treatment failure but to address persistent cavitary lesions, which may limit the generalizability of the findings to the entire spectrum of TB disease. In countries with a high prevalence of MDR-TB, adjunctive surgical resection is a common therapeutic tool which, despite being a major invasive procedure, reduces the transmission burden of MDR-TB and results in favourable outcomes for the patients[59]. However, this approach is uncommon in most countries, thus our results help to understand TB host response but may have a direct impact on TB treatment at short term only in high burden countries where resection is practised. Furthermore, given that individuals with TB may have several lesions at varying stages, which can evolve and recede (as shown in experimental animal studies[2,5,60,61]), expanding the sample size to include several lesions from the same individuals would be beneficial. However, achieving this is practically unfeasible without conducting a complete pulmonectomy or lung section resection. Consequently, working with samples collected post-mortem could offer a viable solution, offering substantial insights and information in this regard, although this is limited by the number of TB patients from whom post-mortem samples would be available and the quality of the samples collected. Another limitation is that given the source and status of the human TB lesion samples, it was required to lower the RINe cut-off to four, acknowledging this a potential bias in RNA-seq experiments. Finally, although we used uninvolved lung parenchyma from our cohort participants as controls, this approach does not eliminate the possibility that immunological influences from the TB lesion environment could affect these uninvolved areas, potentially biasing our results. Nevertheless, our findings clearly distinguish non-lesional tissue from TB lesion tissue, particularly within the central and internal compartments.

In conclusion, we have defined a robust signature for human advanced TB lesions, despite the inter-lesion heterogeneity. Moreover, this is a study showing different modular transcriptomic signature patterns, integrating and co-analysing our findings with TB patients' clinical/microbiological characteristics, including severity and response to treatment. Our study provides a considerable dataset on TB lesions gene expression which will undoubtedly be of broad utility, interest and significance to the scientific community, contributing to an increase in knowledge on TB immunopathology. A better understanding of disease processes and host protective immune responses may help in the clinical management of TB and development of treatment strategies. Most importantly, our findings provide evidence of the clinical picture with a relationship between clinical parameters, treatment response and immune signatures at the infection site.

## Methods

### Ethics

This study is part of the SH-TBL project (ClinicalTrials.gov Identifier: NCT02715271). The protocol, research methodology and all associated documents (informed consent sheet, informed consent form) were reviewed and approved by both ethics' committees at the National Center of Tuberculosis and Lung Diseases (NCTLD) (IRB00007705 NCTLD Georgia #1, IORG0006411) and the Germans Trias i Pujol University Hospital (EC: PI-16-171). Written informed consent was obtained from all study participants before enrolment.

### Study design and patient cohort

The 14 individuals (7 males and 7 females) included in this project were recruited from the SH-TBL cohort, a cross-sectional study conducted at the National Center for Tuberculosis and Lung Diseases (NCTLD) in Tbilisi, Georgia, from May 2016 to May 2018. This study enrolled 40 adult patients who had received an indication for therapeutic surgery for pulmonary TB (ClinicalTrials.gov NCT02715271). All volunteers received standard anti-TB treatment (ATT) regimen according to Georgia national guidelines, and were microbiologically cured, as per WHO definition. Patients were indicated for surgery due to persistent radiological signs of cavitary lesions on Chest X-Ray (CXR) and computed tomography scan, disregarding the drug-sensitivity pattern of the strain responsible and following the official Georgian National Guideline "Surgical Treatment of Patients with Pulmonary Tuberculosis"[59]. Thoracic surgery decisions were made by the NCTLD Tuberculosis Treatment Committee, composed of two surgeons and 18 pulmonary TB specialists.

### Data and sample collection

Anonymised data regarding the socio-epidemiological factors, clinical aspects, and information referring to the current TB episode for the SH-TBL cohort were collected using an electronic case report form. Data available were: demographic (self-reported biological sex, age); clinical data (BMI, presence of symptoms assessed using SGRQ symptom sub-score to evaluate the frequency and severity of key respiratory symptoms, C-Reactive Protein (CRP) value); data on TB episode (relapse or new TB case); microbiological data (drug sensitivity, SCC); radiological data (number of lesions in CXR and lesion localization within the lung); data on resected TB lesion and pathology analysis data (Supplementary Data 2).

During surgery, TB lesions were removed (median of 3.2 cm in diameter), and cut to obtain: (1) one piece containing all compartments and non-lesional tissue for pathology studies; and (2) 48 biopsy fragments (-0.5 cm³) of tissue samples in RNA*later* solution (Qiagen) at 4 °C overnight, before storage at −80 °C for further RNA-Seq analysis. These biopsy fragments were collected from each differentiated zones of the TB lesion by macroscopic examination by a pathologist: Central Lesion (C), Internal Wall (I), External Wall (E). In addition, surrounding non-lesional (NL) lung parenchyma tissue, unaffected, by eye and by palpation, was collected from the same patient (Fig. 2a). Samples were processed in BioSafety Level 3 (BSL-3) laboratory.

## Total RNA extraction

For an optimal RNA recovery, TB lesion biopsy samples were divided into 0.16–0.21 g single pieces and placed into new tubes. Samples were reduced to powder by mechanical cryofracturing using a BioPulverizer device (Biospec Products) after being cooled in liquid nitrogen. The powdered tissue was then transferred to 2 mL Lysing Matrix D tubes together with lysis solution for homogenization by FastPrep® instrument (MP Biomedicals). RNA was purified using the mirVana miRNA Isolation Kit (Thermo Fisher Scientific, AM1560), followed by genomic DNA digestion using the DNA-*free* DNA Removal Kit (Thermo Fisher Scientific, AM1906) according to manufacturer's instructions. Quantitative and qualitative RNA integrity number equivalent (RINe) values were obtained by Agilent Bioanalyzer 2100 (Agilent Technologies). In general, a standard RINe score for good quality RNA is set at seven. Considering the source and status of the human TB lesion samples, and our samples ranging from 4 to 7.4 (Supplementary Data 2), a minimum RINe cut-off of four was established.

## RNA-Sequencing library preparation, sequencing, and gene alignment

Purified RNA was diluted to 25 ng/μl per aliquot and then shipped on dry ice to Macrogen (Seoul, South Korea), where the RNA-sequencing (RNA-Seq) was performed. Libraries were constructed using the TruSeq Stranded Total RNA LT Sample Prep Kit (Human Mouse Rat) (Illumina, RS-122-220X) following the TruSeq Stranded Total RNA Sample Prep guide (Part #15031048 Rev. E), including prior removal of ribosomal RNA using the RNA Ribo-Zero rRNA Removal Kit (Human/Mouse/Rat) (Illumina). RNA-Seq was performed on an Illumina platform HiSeq 4000 (Illumina), at 50 million reads per sample, 100 bp stranded paired-end reads. Pre-processing of raw data included quality control through FastQC (v.011.7) and MultiQC (v.1.9)[62]. Before further steps in read pre-processing, Illumina adapters were trimmed off with Trimmomatic (v.0.39)[63]. The human genome sequence GRCh38.89 and human gene annotations were downloaded from the ENSEMBL web repository. Files from each sample were aligned to the human reference genome using the Spliced Transcripts Alignment to a Reference (STAR) package (v.2.7.5b)[64], with the built-in gene counts quantification mode for stranded RNA-Seq data. BAM files were generated, and the SAMtools package applied to calculate the percentage of successful read alignment against the reference human genome (v.1.10)[64].

## RNA-seq data analysis

The overall pipeline for data handling, plotting and statistical analysis was conducted in R (v.4.3.3). After STAR mapping, a gene count data table was obtained including C, I, E and NL samples. Genes with a lower than 50 counts among all the samples were discarded to avoid confounding the differential gene expression analysis, as they had low expression to be reliably quantified. Paired statistical analyses were done globally and separately for each compartment. The set for the RNA-Seq experiments comprised 48 samples from 14 patients. Samples from patient SH-TBL03 weren't taken into consideration for the paired comparisons between the whole TB lesion and separated compartments as the NL tissue sample control was missing. The differential expression analysis from tissue count tables was conducted using the DESeq2 Bioconductor package (v.1.28.1)[65]. Genes were considered as significant DEGs when the Benjamini–Hochberg adjusted $p$-value was equal to or less than 0.05 ($p \le 0.05$). The R package heatmap (v.1.0.12) was used to generate heatmaps and dendrograms for the genes and samples by hierarchical clustering after DESeq2 depth normalization. Heatmaps describe the Euclidean distances between samples.

## Enrichment score for the different tissue compartments

The expression across compartments of upregulated selected genes differentiating granuloma from non-lesional tissue was performed using ssGSEA. ssGSEA is a variation of the GSEA algorithm that instead of calculating enrichment scores for groups of samples and sets of genes, it provides a score for each sample and gene set pair[66].

## Weighted gene co-expression network analysis and functional annotation

Weighted gene co-expression network analysis (WGCNA) was performed to identify modules using the R package WGCNA (v.1.72-1). The TB granuloma modules were constructed using the 10,000 most variable genes across all TB samples collected (log2 RNA-seq expression values). To satisfy the scale-free topology criteria and the recommendations for WGCNA use, we chose an optimal soft-threshold ($\beta = 23$) to obtain an adjacency matrix from a signed weighted correlation matrix containing pairwise Pearson correlations, generating the corresponding topological overlap measure. To detect the modules, we applied a dynamic hybrid tree-cut algorithm to detect the computed modules of co-expressed genes (minimum module size of 20, and deep split = 1). Module colours represent distinct clusters of genes that are grouped together based on similarity in their expression profiles. Finally, 21 modules were obtained. An additional "grey" module was identified in TB granuloma modules, consisting of genes that were not co-expressed with any other genes. The grey module was discarded from further analysis. Moreover, only modules with more than 40 genes were annotated. We computed their intramodular connectivity and selected the top five most interconnected genes[67]. Significantly enriched Gene Ontology and canonical pathways from the MSigDB website[68] were computed using clusterProfiler R package[69]. Modules were annotated based on representative biological processes from pathways and processes from all three reference databases. Fold enrichment for the WGCNA modules was calculated using the quantitative set analysis for gene expression with the Bioconductor package QuSAGE[70]. To identify the modules of genes over or underabundant in TB granuloma, compared to the respective non-lesional lung tissue using log2 expression values using the three compartments. Only modules with enrichment scores with FDR $p$-value < 0.1 were considered significant.

## Sub-modules representative of immune populations analysis

The LM22 immune population transcriptomic signature[21] was used to know the location of genes associated to immune populations within the WGCNA modules, while the CIBERSORT deconvolution function from the IOBR package[71] was used to estimate the proportions of the LM22 populations across TB lesion compartments. The two largest immune-associated modules with the highest number of genes, the adaptive/humoral and innate/PRR modules, were then expanded by retrieving the unmerged modules contained within them before the module merging step from the WGCNA analysis. The clusterProfiler R package[69] was used to compute the significantly enriched Gene Ontology terms and annotate the sub-modules. Fold enrichment was calculated using the quantitative set analysis for gene expression with the Bioconductor package QuSAGE, employing the three compartments compared to non-lesional tissue as before and considering as significant an FDR < 0.1.

## Association between modules and clinical characteristics

TB individuals were classified and categorised taking into consideration clinical surrogates of disease severity, using the following parameters: SGRQ symptoms sub-score > 20 or <20; being a fast (SCC < 2 months) or slow sputum culture converter (SCC > 2 months) after the start of ATT); DS vs MDR-TB case; being a relapse or new TB case; number of lesions present in the CXR. The SGRQ symptoms score comes from an eight-item questionnaire with a weighted score ranging from 0 to 100, with higher scores indicating higher effects, frequency and severity of respiratory symptoms (Supplementary Data 2). To divide the patients for the analysis hereby presented we used a cut-off defined by the median SGRQ symptoms value, >20 being considered more severe.

We computed the eigengene for each module, defined as the first principal component of the module representing the overall expression level of the module. The relationship of the transcriptomic modules with clinical surrogates of TB severity (SCC and SGRQ symptoms score) was tested using Wilcoxon-rank sum test. Nominal *p*-values were adjusted using the Benjamini–Hochberg approach[72].

## Identification of transcription factors

Transcription factors were identified from the list of differentially expressed genes between TB lesion (G) vs NL filtered by genes belonging to modules associated with TB severity (SCC and SGRQ symptoms score) by using the BioMart R package[73] and filtering by the Gene Ontology term GO:00037000, which corresponds to DNA-binding transcription factor activity[74]. This yielded a list of 92 transcription factors which was then tested using the Wilcoxon-rank sum test. Nominal *p*-values were adjusted using the Benjamini–Hochberg approach[72].

## Immunohistochemistry validation

The module with the highest enrichment within the TB lesion and association to TB severity (SCC and SGRQ symptoms score), the IFN/cytokine signalling module, was used to generate a protein-protein interaction (PPI) network with the STRING 12.0 website[75] set at the highest confidence interaction score (<0.9). The CytoHubba plugin for Cytoscape 3.10.2 was used to identify the top 5 hub genes with 12 topological analysis methods. The three genes that were selected the most by the 12 methods and which overlapped with the top 10 hub genes ranked by module membership from the IFN/cytokine signalling module were selected for immunohistochemistry validation. Immunohistochemistry analysis was carried on paraffined sections of lesion samples and lung samples from individuals who had undergone surgery for bullous emphysema (as non-TB controls). The staining was performed with antibodies against CXCL9 (rabbit polyclonal 22355-1-AP; Proteintech, dilution 1:100), GBP5 (rabbit polyclonal 13220-1-AP; Proteintech, dilution 1:200), and STAT1 (rabbit polyclonal 10144-2-AP; Proteintech, dilution 1:300). Slides were scanned with an AxioScan 7 and Zen 3.10 imaging software (Zeiss). Image analysis was performed using ImageJ and applying colour deconvolution for haematoxylin and DAB. Thresholding was used to quantify the stained areas, with thresholds set from 0 to 175 for haematoxylin and 0 to 140 for DAB. Total stained area was measured for each colour and the results were expressed as a percentage of DAB staining per total tissue area. Statistical analysis was performed by applying the Wilcoxon-rank sum test and adjusting nominal *p*-values with the Benjamini–Hochberg approach[72].

## Reporting summary

Further information on research design is available in the Nature Portfolio Reporting Summary linked to this article.

## Data availability

The metadata and sequencing data generated in this study have been deposited in the National Center for Biotechnology Information Gene Expression Omnibus (GEO) database under accession code GSE184537. The remaining data generated in this study are provided within the Article, Supplementary Information, Supplementary Datas 1 and 2, and Source Data file or from the corresponding author on request. These patient data used in this study are freely available in the Mendeley database under the following (https://doi.org/10.17632/knhvdbjv3r.1)[76]. Source data are provided with this paper.

## Code availability

The code used in this study is available on GitHub, https://doi.org/10.5281/zenodo.15322541[77].

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

## Acknowledgements

The authors would like to thank the individuals with TB who agreed to participate in the study and the staff from the NCTLD who helped with the SH-TBL project. The authors thank the support from Comparative Medicine and Bioimage Centre of Catalonia (CMCiB), namely at BSL 3 facility, Eric Garcia for technical support in acquiring the data. The authors, thank William J. Branchett, from The Francis Crick Institute, London, for his input on the manuscript and providing suggestions/changes for improvement. The authors thank the Pathological Anatomy department of the HUGTIP and Joaquim Grego from the Advanced Microscopy Unit of the Josep Carreras Leukaemia Research Institute for the immunohistochemistry processing and imaging. This work was supported by: (1) the Plan Nacional I + D + I co-financed by ISCIII-Subdirección General de Evaluación and Fondo-EU de Desarrollo Regional (FEDER) through PI16/01511 (CV), PI20/01424 (CV), CP13/00174 (CV), CPII18/00031 (CV), RYC-2010-07249 (CA), CB06/06/0031 (PJC), CB06/04/0033 (CA), (2) The European Union's Horizon 2020 research and innovation program under grant agreement No 847762 (CV). (3) The Catalan Agency for Management of University and Research Grants (AGAUR) through 2017SGR500 (PJC), 2021 SGR 00920 (CV), 2017-SGR-490 (CA), 2021 SGR-01186 (CA) and 2017 FI_B_00797 (AD), 2022_FI_B00528 (AdRA) and 2019 FI_B01024 (JCR). (4) The "Spanish Society of Pneumology and Thoracic Surgery" (SEPAR) (16/023) (CV). (5) The Wellcome Trust, the Medical Research Foundation grants (206508/Z/17/Z and MRF-160-0008-ELP-KAFO-C0801) (MK) and the NIHR Imperial College BRC (MK). (6) Wellcome Trust (204538/Z/16/Z) and National Centre for the Replacement, Refinement and Reduction of Animals in Research (NC3Rs) (NC/R001669/1) grants (SJW) (JCR). (9) The Francis Crick Institute (AOG) receives its core funding from Cancer Research UK (FC001126), the UK Medical Research Council (FC001126), and the Wellcome Trust (FC001126); before that by the UK Medical Research Council (MRC U117565642).

## Author contributions

Conception: C.V. and S.V. Design of the work: C.V. and S.V. with substantial contributions of M.K. and A.O.G. K.L.F., A.D., J.J.L., D.A., J.S., L.A., D.H.C., A.d.R.A., J.C.R., A.G., L.M.W., G.T., P.R.M., P.J.C., S.V., S.G., K.N., N.S. and Z.A. worked on the acquisition and analysis of data for the work, and all authors made substantial contributions to its interpretation. RNA-Sequencing library preparation, sequencing, gene alignment and initial bioinformatics analysis was performed by A.D. and D.H.C., supervised by M.K.; and paired statistical analysis, single sample Gene Set Enrichment analysis, WGCNA analysis and the association between modules and clinical characteristics were performed by K.L.F., D.A. and J.J.L., supervised by A.O.G. G.T. and P.R.M. performed the pathological analysis and immunohistochemistry staining. K.L.F., D.A. and C.V. drafted the work; and C.A., F.M.T., A.S., A.G.C., S.V., S.J.W., M.K. and A.O.G. reviewed it critically for important intellectual content. All authors revised and gave their final approval of the version to be published and agreed to be accountable for all aspects of the work in ensuring that questions related to the accuracy or integrity of any part of the work are appropriately investigated and resolved.

## Competing interests

The authors declare the following competing interests: C.V. is an unpaid board member of the following non-profit organizations: the FUITB foundation and the ACTMON foundation. Neither the FUITB nor ACTMON have had any role in the conceptualization, design, data collection, analysis, decision to publish, or preparation of the manuscript. The remaining authors declare no competing interests.

## Additional information

[1]Experimental Tuberculosis Unit (UTE), Fundació Institut Germans Trias i Pujol (IGTP), Badalona, Spain. [2]Centro de Investigación Biomédica en Red de Enfermedades Respiratorias (CIBERES), Madrid, Spain. [3]Bioinformatic Platform, Centro de Investigación Biomédica en Red de Enfermedades Hepática y Digestivas (CIBERehd), Barcelona, Spain. [4]Hospital Clínic de Barcelona, Barcelona, Spain. [5]Department of Infectious Diseases, Imperial College London, London, UK. [6]Department of Genetics and Microbiology, Universitat Autònoma de Barcelona (UAB), Bellaterra, Spain. [7]Childhood Liver Oncology Group, Germans Trias i Pujol Research Institute (IGTP), Badalona, Spain. [8]Translational Program in Cancer Research (CARE), Germans Trias i Pujol Research Institute (IGTP), Badalona, Spain. [9]Liver and Digestive Diseases Networking Biomedical Research Centre (CIBER), Madrid, Spain. [10]Global Health and Infection, Brighton and Sussex Medical School, University of Sussex, Brighton, UK. [11]National Center for Tuberculosis and Lung Diseases (NCTLD), Tbilisi, Georgia. [12]European University, Tbilisi, Georgia. [13]Pathology Department, Hospital Germans Trias i Pujol, Badalona, Spain. [14]Universitat Autònoma de Barcelona, Barcelona, Spain. [15]Microbiology Department, Northern Metropolitan Clinical Laboratory, Hospital Universitari Germans Trias i Pujol, Badalona, Spain. [16]Genetics, Vaccines and Pediatric Infectious Diseases Research Group (GENVIP), Instituto de Investigación Sanitaria de Santiago (IDIS), Santiago de Compostela, Spain. [17]Universidad de Santiago de Compostela (USC), Santiago de Compostela, Spain. [18]Translational Pediatrics and Infectious Diseases, Hospital Clínico Universitario de Santiago de Compostela (SERGAS), Santiago de Compostela, Spain. [19]Unidade de Xenética, Instituto de Ciencias Forenses (INCIFOR), Universidade de Santiago de Compostela, Santiago de Compostela, Spain. [20]GenPoB Research Group, Instituto de Investigación Sanitaria (IDIS), Santiago de Compostela, Spain. [21]Centre for Paediatrics and Child Health, Imperial College London, London, UK. [22]Immunoregulation and Infection Laboratory, The Francis Crick Institute, London, UK. [23]National Heart and Lung Institute, Imperial College, London, UK. [24]The University of Georgia, Tbilisi, Georgia. [25]These authors jointly supervised this work: Sergo Vashakidze, Cristina Vilaplana ✉e-mail: cvilaplana@igtp.cat

