## [Transparent Peer Review file · Nature Communications]

Unravelling the transcriptome of the human tuberculosis lesion and its clinical implications

Corresponding Author: Dr Cristina Vilaplana

Version 0:

Reviewer comments:

Reviewer #1

(Remarks to the Author)

In this study the authors characterized 44 fresh human pulmonary TB lesion samples from 13 patients undergoing therapeutic surgery. Samples were obtained from drug-sensitive and multi-drug resistant TB. Total RNA from three spatially distinct sections: Central Lesion, Internal Wall and External Wall were collected from each patient's granuloma biopsy and gene expression across these spatially distinct granuloma regions was evaluated by bulk RNA sequencing. A clear demarcation with increased inflammatory profile was observed between the granuloma and adjacent non-lesional tissue. Using weighted correlation network analysis, the authors found a significant enrichment of transcriptional modules related to inflammation across all compartments. Furthermore, granuloma modular transcriptional signature correlated with the patients' clinical and microbiological status with increased IFN/cytokine signaling and neutrophil degranulation modules observed in patients who were slow sputum converters and those with more disease severity. The authors conclude that their findings provide evidence of a relationship between clinical parameters, treatment response and immune signatures at the infection site.

Comments

1. Not clear how the transcriptional findings between the slow and fast SCC converters can inform management of the disease and the potential use of SCC at the two-month mark after treatment initiation as a prognostic indicator since all patients were critically ill and had to have lung resection surgery.
2. The data are more confirmatory in nature of previously published work in humans and animal models. The increase in certain transcriptional modules such as IFN/cytokine signaling and neutrophil degranulation modules associating with poorer clinical response is not entirely surprising. The transcriptional findings do not provide mechanistic insights into why there is difference in time to sputum conversion in patients with fast and slow SSC and those with differences in the SGRQ symptom score.
3. It would have been helpful to show immunohistochemistry confirmation of RNA-Seq data and also correlation to bacterial burden at the different spatial locations.

(Remarks on code availability)

Reviewer #2

(Remarks to the Author)

The authors analyze the transcriptional signature of different regions of the TB granuloma in a large number of human samples. They identify neutrophil degranulation and IFN/cytokine signaling as two of the most striking modules increased in the granuloma compared to non-lesion tissue. They also show a connection between sputum culture conversion & symptoms score with the granuloma signature, confirming that tissue events are reflected in the systemic outcome of disease. Methods are very detailed and easy to follow.

Major comments:

In line 151-153 it is mentioned that 7 granuloma cluster together with non lesion tissue based on top 40 DEGs. This is very interesting but not elaborated on further. Are these granuloma all derived from the same patient with less severe disease? Or are these all external wall samples? The manuscript focuses on the relationship between granuloma transcriptome and clinical status, so I feel these outliers should be discussed in more detail.

Figure 3 focuses on the analysis of the separate granuloma areas, but all comparisons are performed against non-lesional tissue. It would be relevant to also specifically identify differences between the different granuloma sections, particularly core and internal wall. It is possible to see some of these differences in figure 3c but a direct comparison would give more insight into these differences, and this can then also be stratified against the clinical data. This may reveal that patients with less severe disease scores have a reduced core transcriptional activity, or a particular cellular signature in the internal wall indicative of more or less immune activation. Performing this analysis to maximize the information that can be gained from the heterogeneity of the granuloma. This may also add to the novelty of the study as many of the signatures identified have been previously published.

Minor comments:

Line 156 - remove diverse, as diverse and diversity are both mentioned in the sentence, doubling up

Line 179 - significantly should be significant

It is unclear what the module colours mentioned in the figures add, please explain or remove.

(Remarks on code availability)

Reviewing this code is outside my expertise

Reviewer #3

(Remarks to the Author)

This study explores the transcriptional profile of diverse lesions obtained from n=13 patients with pulmonary TB. Bulk RNA-seq of multiple tissue biopsies (n=44) obtained from different sites of TB lesions was used along with weighted correlation network analysis to map diverse microenvironments in the Mtb-infected lungs. The transcriptional profiles were associated to disease severity and clinical outcome. The main findings presented were that more severe lesions displayed enhanced inflammation characterised by IFN-signalling and neutrophil degranulation.

Although I sincerely understand and appreciate the efforts invested into this precious tissue material including the impressive RNA analyses, the significant advances obtained from this study are not clear. Enhanced inflammation in severe TB disease along the IFN/neutrophil axis is not new. Instead, the manuscript is entirely void of data describing T cell responses in these different granuloma sites. A significantly more detailed description of the outline and analyses in the MM section is required to understand the specific clinical and pathological criteria used in the study. In addition, the manuscript would benefit from adding some type of spatial image analysis (at the protein level) including cells and markers that could support the RNA-seq data, which would bring significant added value to the study.

Specific comments:

1. While the RNA-seq analysis done is extensive it is also the sole experimental pillar in this study. As multiple tissue biopsies were collected from each patient, the authors should present complementary data on cell types (eg. macrophages, T and B cells, neutrophils, epithelial cells, fibroblasts) as well as protein expression and distribution in the tissue lesions using multiplex imaging or similar technology. This would bring significant added value to the RNA-seq analyses.

2. Have a control group been included eg. representing uninfected individuals who undergo thoracic surgery due to another disease indication such as lung cancer? This would add relevance for comparison to the RNA-seq analysis, and also to the demographic data where for example BMI is relatively high among the TB patients. Why is that? Wasting and weight loss is one of the primary clinical symptoms of chronic TB, while the 14 patients in this study are normal weight or obese according to Table S1. This should be explained and discussed and preferentially related to a control cohort.

Additionally, were blood chemistry analyses performed that could provide additional information on disease severity eg. ESR/CRP, WBC, CD4/CD8 T cell counts, Hb, albumin?

3. The Introduction seems to be written in a similar way as compared to the Discussion. I lack a clear introduction and background that clearly describes the aims of the study and that provides a solid background to current knowledge about granuloma composition and structure. Three initial references in the Introduction refer to WHO; is this necessary? Currently, the authors put a lot of focus into citing recent high-profile studies in the field, while I miss out on several original references on TB granuloma imaging and computerized quantification of cells and molecules both the RNA and protein level, such as PMID: 10768979, 11918694, 12379712, 14638800, 17664265, 25510482, 19435796, 19435796, 15376257, 35092727, 31015452, 33552077. This is particularly important as many of these references refer to cellular immunity including T cell responses that are largely missing from the current discussion.

4. As the patients were selected for adjunct surgery based on their chest X-ray findings and response to treatment, have any type of radiological scoring been used to quantify the extent and severity of pulmonary involvement including the extent and character of the inflammatory lesions, consolidations, level of fibrosis and cavity formation (size and numbers, lung lobe and

zone)? There are multiple manual scoring systems available in addition to computer-aided detection of TB. Chest X-ray grading can also be done retrospectively on existing X-rays. Currently, the only data available is “multiple lesions in the CXR less or equal to/more than 2”, which provides very limited information.

5. How come n=7 DS-TB patients were referred to surgery or what were the specific criteria or indication for surgery? With what method was drug susceptibility of the Mtb isolates tested? A Table or description of the drug-resistance present should be added.

6. In the MM section it is described that the St. George's Respiratory Questionnaire (SGRQ) was used to assess the presence of TB symptoms. How come the authors used this score that does not seem to be specifically designed to measure TB symptoms, when there are validated composite TB scores available (eg. PMID: 17852907, 23113626, 33757875, 24857613)? The variables included in the SGRQ score should be listed and explained in the manuscript. Currently, there is no information as to what components (relevant to TB disease) this score contains or what a threshold of 20 means (as the score ranges from 0 to 100, correct?).

It is also stated that: “TB individuals were classified taking into consideration clinical surrogates of disease severity.” Meaning what? SCC and SGRQ symptom score? Was there a difference in the SGRQ score comparing DS- to DR-TB? Was there a correlation between time to sputum culture conversion and SGRQ?

7. It is unclear for the MM section exactly how the different lesion areas were selected and scored. Only by macroscopic examination by a pathologist? Was HE-stains made to support macroscopic findings? Pls, define “granuloma sample”. Currently, only the size of the lung biopsies is given, but how can one single granuloma be detected and dissected from each piece of tissue as confluent granulomas of different sizes are very common?

In Table S1. “Fresh vs Not fresh necrosis” does not make sense to me. It is confusing to place parts of the characteristics for the TB lesions in Table S1, and another part of characteristics of TB lesions according to treatment response as Table S2. An explanation to the “category” Cavitation and Tuberculoma is also required.

8. In Fig 2a, the TB granuloma is depicted in the lower part of the lung, while lesion is normally found in the apical parts of the lung. Please, also specify which lung segments/lobes the biopsy material was obtained from. In Fig 2a, it would also be helpful to list, for each segment, C, I, E or NL, the criteria/features used for dissecting each tissue section into a separate vial for the RNA analyses. In addition, the colour scheme becomes a bit too much with all the colours depicted, eg. “Granuloma” could perhaps be described in plain black or grey text (and colour in Fig 2b).

Pls, list the individual RIN values for each tissue specimen in Supplementary Data set 2. I cannot find any information on gene expression modules including T cell responses in any of the tissue specimen analysed, why is that?

9. The Discussion could be rewritten to include a more integrated mechanistic explanation on the humoral response, DNA binding and myeloid activation modules together with the IFN and neutrophil modules detected in the lesions. Nothing is mentioned about T cell responses that are known to be crucial for Mtb control, can the authors please comment on that? Overall, the results are presented broadly now, and the manuscript could benefit from some focus that ties all the loose ends together, perhaps also using a graphical illustration of the main conclusions.

In several places in the Discussion, the authors indicate that responses observed in peripheral blood reflect the microenvironment at the site of infection. This is debatable, and I would strongly recommend lifting the advantages of studying responses at the granuloma sites.

10. Minor comment: The manuscript needs language editing. For example, second sentence in the abstract “... 13 patients undergoing therapeutic surgery using RNA-Sequencing.” is not correct. Overall, there are numerous text parts and sentences throughout that appear misplaced and disturb the reading flow of the paper.

(Remarks on code availability)

Version 1:

Reviewer comments:

Reviewer #1

(Remarks to the Author)

The authors have not completely addressed my concern regarding the usefulness of comparing the RNA-seq data to sputum conversion and disease severity given that all TB individuals recruited to the study underwent therapeutic surgery and must have presented with severe immunopathological disease. How will the findings inform the treatment plan? A modified discussion to address this would be helpful.

(Remarks on code availability)

Reviewer #2

(Remarks to the Author)

I am satisfied with the changes made by the authors in response to my comments and think the revised manuscript is suitable for publication.

(Remarks on code availability)

Reviewer #3

(Remarks to the Author)

I believe the authors have addressed most of my concerns and significantly improved the manuscript, particularly by including several clarifications in the Materials and Methods section, performing immunohistochemistry (IHC) on selected proteins, and utilizing the LM22 immune population transcriptomic signature to profile human hematopoietic cell populations in the lesions. However, it has also become clear that the patients included in this study were not subjected to surgery due to treatment failure but were considered microbiologically cured. This is important information regarding their clinical TB disease and should be further emphasized in the manuscript.

Specific comments (with regards to the Marked copy of the manuscript):

Even though the patient selection, clinical data collection and biopsy sampling are clearer now, I suggest adding a sentence in the first paragraph of the Results that a normal to high BMI, low CRP and relatively low SGRQ score, were consistent with microbiological cure and that these patients were not considered critically ill. I believe this is important information as the readers may otherwise be misguided.

Include data in the MM section from line 723, indicating that the Saint George's Respiratory Questionnaire (SGRQ) symptom scores range from 0 to 100, with higher scores indicating more respiratory limitations. If there is no available information on this score, what is assessed and how it is ranged, it is not possible to critically understand an SGRQ score below or above 20. I also recommend listing the symptoms components of the SGRQ for each patient that was used in this study as a Supplementary data set (perhaps as part of S. data set 2).

It is unclear why the authors omitted lines 270-276? Without this information in the Results, it will be difficult for the readers to follow.

Thank you for adding the RIN values in S. data set 2. I do understand that a lower RIN is to be expected in this type of tissue material, but a RIN value of 4-7.5 is generally considered moderate and may be acceptable for some RNA analyses, but it is not ideal for highly sensitive applications like RNA-seq where a RIN < 8 is typically recommended. Please, briefly comment on this limitation in the manuscript and add information on the RIN values in the MM section of the main manuscript to clarify that the RIN in the different biopsy samples ranged from 4-7.4.

It could assist the readers if the authors could highlight the regions depicted in the images included in the new Fig. 5, eg. necrotic core (NC), lymphocyte rim, alveolar space etc.

I think that the addition of the data on specific immune responses using the LM22 signature matrix is interesting and beneficial to the data processing and presentation. Would it also be possible to use the LM22 transcriptomic matrix to plot the relative percentage of each immune cell subset plotted in Supplementary Figure 3, comparing the relative immune cell distribution in the TB lesion vs non-lesional tissue and an also external wall to central wall? These data could be presented eg. in an overlay bar chart including all immune cell types (Y-axis set to 100%). I believe such data would be informative and complement the data currently presented in S. Figure 3.

In S. Figure 4, I find it interesting that the humoral module appears mostly upregulated, even compared to IFN-signaling. This should be highlighted in the Results of the manuscript (starting from line 245). Could the authors also plot Turquoise 4 and 7 comparing the symptoms score (similar to S. Fig. 4b) as well as comparing fast vs slow SCC? Could this mean that stronger B cell activation including an antibody response in the lesions is associated with a dysregulated immune response locally in the lesions? Perhaps there is a lot of extracellular bacilli in the lesions of these "microbiologically cured" patients that drive inflammation and particularly B cell activation?

Lines 355-358 in the Discussion: How can these results be used to refine anti-TB treatment? Using HDT or something else? Please, briefly develop/clarify.

Line 444 in the Discussion: I realize the module data shown in S. Figure 4a, supports the assumption that T cells are enriched in the central and internal compartment of the TB lesions. However, most literature on human TB granulomas, including studies from the groups of JL Flynn, G Kaplan, and T Ulrichs etc, features lymphocytes in the peripheral rim of the granuloma. These lymphocytes surround Mtb-infected macrophages, possibly preventing the effective killing of infected cells. Can the authors comments on this and adjust their Discussion accordingly?

Finally, language, spelling and abbreviations needs to be checked (some examples provided below):

Line 300: strange order of words

Line 742: sentence is odd

Line 720: incorrect?

S. Fig. 4b: upper left graphs, SQRQ instead of SGRQ

Reply to question 1: incorrect Supplementary Figures 4 and 5a-b are referred to, perhaps authors mean S. Figure 3 and 4?

(Remarks on code availability)

Version 2:

Reviewer comments:

Reviewer #1

(Remarks to the Author)

My concerns have been addressed.

(Remarks on code availability)

Reviewer #3

(Remarks to the Author)

I thank the authors for the revised version of the manuscript and can confirm that all my remaining concerns have been appropriately addressed.

(Remarks on code availability)

ANSWERS TO REVIEWERS

ANSWERS TO REVIEWER #1

Dear reviewer, please find here below our answers to your queries and suggestions in red, and changes to the figures highlighted in yellow.

Many thanks in advance,

Yours sincerely,

The authors

In this study the authors characterized 44 fresh human pulmonary TB lesion samples from 13 patients undergoing therapeutic surgery. Samples were obtained from drug-sensitive and multi-drug resistant TB. Total RNA from three spatially distinct sections: Central Lesion, Internal Wall and External Wall were collected from each patient's granuloma biopsy and gene expression across these spatially distinct granuloma regions was evaluated by bulk RNA sequencing. A clear demarcation with increased inflammatory profile was observed between the granuloma and adjacent non-lesional tissue. Using weighted correlation network analysis, the authors found a significant enrichment of transcriptional modules related to inflammation across all compartments. Furthermore, granuloma modular transcriptional signature correlated with the patients' clinical and microbiological status with increased IFN/cytokine signalling and neutrophil degranulation modules observed in patients who were slow sputum converters and those with more disease severity. The authors conclude that their findings provide evidence of a relationship between clinical parameters, treatment response and immune signatures at the infection site.

Comments

1. Not clear how the transcriptional findings between the slow and fast SCC converters can inform management of the disease and the potential use of SCC at the two-month mark after treatment initiation as a prognostic indicator since all patients were critically ill and had to have lung resection surgery.

Answer: The reviewer is correct; all patients in this study presented with lesions on their X-rays, despite receiving appropriate treatment and achieving microorganism clearance from sputum. This ultimately led to the need for surgery, though none were critically ill. To clarify this, we have removed "from critically ill patients" from the discussion section (line 388 of the tracked-changes version provided). In our view, establishing a link between a slower culture conversion rate (persisting beyond 8 weeks post-treatment initiation) and the presence of more inflamed lesions at treatment's end opens the potential for refining TB treatment during clinical management. To clarify this, we have reformulated the sentence to emphasize this point in the Discussion section at lines 357-360, and included a new figure (Fig 6.), as suggested by reviewer 3, summarizing the main conclusions of the paper.

2. The data are more confirmatory in nature of previously published work in humans and animal models. The increase in certain transcriptional modules such as IFN/cytokine signalling and neutrophil degranulation modules associating with poorer clinical response is not entirely surprising. The transcriptional findings do not provide mechanistic insights into why there is difference in time to sputum conversion in patients with fast and slow SSC and those with differences in the SGRQ symptom score.

Answer: To address the reviewer's comment, we have conducted further analysis to gain insight into the differences between patients' clinical surrogates, obtaining a set of transcription factors that are differentially expressed between fast and slow sputum culture converters and patients with less severe or more severe symptoms. As a result, we have modified the manuscript text in the following sections: in Results section (lines 307-314), in Discussion Section (449-458, 483-493) and in Methods Section (lines 739-746). We have also added a new Supplementary Table 2 and Supplementary Figure 6.

3.1 It would have been helpful to show immunohistochemistry confirmation of RNA-Seq data.

Answer: Following the reviewer's suggestion, we have conducted immunohistochemistry confirmation of RNA sequencing data from three genes (*CXCL9*, *GPB5* and *STAT1*) significantly expressed between granuloma and NL tissue, and that corresponds to the module with the highest enrichment associated with TB severity (SCC and SGRQ symptoms score). As a result, we have modified the manuscript text in the following sections: in the Results section (lines 315-321), in Discussion section (lines 493-496) and in Methods Section (lines 748-769). We have also added Figure 5.

3.2 It would have been helpful to also show correlation to bacterial burden at the different spatial locations.

Answer: We agree with the reviewer's suggestion and understand the importance of searching for associations between the bacterial burden in the different compartments and the RNA sequencing data. Accordingly, we assessed the bacterial presence and absence (using the AFB in the lesion according to the available data) and the bacterial burden (using the ZN grading recorded at the time of surgery), both included in Supplementary Dataset 2 and, for the different compartments with the representative genes (*CXCL9*, *GPB5* and *STAT1*) used in the previous point. Unfortunately, no definitive conclusions could be drawn, as stratifying by compartment reduced the sample size per condition. Moreover, 8 samples were dismissed because of the RNA quality (mainly from central compartment). No figure was included as no conclusive and significant data was found to be associated to the bacterial burden in this cohort.

ANSWERS TO REVIEWER #2

Dear reviewer, please find here below our answers to your queries and suggestions in red, and changes to the figures highlighted in yellow.

Many thanks in advance,

Yours sincerely,

The authors

The authors analyze the transcriptional signature of different regions of the TB granuloma in a large number of human samples. They identify neutrophil degranulation and IFN/cytokine signaling as two of the most striking modules increased in the granuloma compared to non-lesion tissue. They also show a connection between sputum culture conversion & symptoms score with the granuloma signature, confirming that tissue events are reflected in the systemic outcome of disease. Methods are very detailed and easy to follow.

Major comments:

In line 151-153 it is mentioned that 7 granuloma cluster together with non-lesion tissue based on top 40 DEGs. This is very interesting but not elaborated on further. Are these granuloma all derived from the same patient with less severe disease? Or are these all external wall samples? The manuscript focuses on the relationship between granuloma transcriptome and clinical status, so I feel these outliers should be discussed in more detail.

Answer: Indeed the 7 granuloma samples clustering with non-lesional tissue included six samples from external compartment samples and one from the internal compartment. Following the reviewer's suggestion, we have revised and included a sentence in the results section detailing the compartments at which the samples belong (Results section lines 175-180).

Figure 3 focuses on the analysis of the separate granuloma areas, but all comparisons are performed against non-lesional tissue. It would be relevant to also specifically identify differences between the different granuloma sections, particularly core and internal wall. It is possible to see some of these differences in figure 3c, but a direct comparison would give more insight into these differences. Then this can then also be stratified against the clinical data. This may reveal that patients with less severe disease scores have a reduced core transcriptional activity, or a particular cellular signature in the internal wall indicative of more or less immune activation. Performing this analysis to maximize the information that can be gained from the heterogeneity of the granuloma. This may also add to the novelty of the study as many of the signatures identified have been previously published.

Answer: Following the reviewer's suggestion, we produced the corresponding heatmaps for the requested comparisons and added them as Supplementary Figure 2 and referenced it in Results section (line 210-211). It is now easier to observe the differences between the central and internal/external granuloma sections; while the magnitude of differential

expression on the internal lesion relative to the external compartment show a gradually decreased towards the edge of the granuloma structures highlighting the transition from central to internal and external areas. This behaviour was previously observed when comparing the external compartment to the NL tissue showing that, with the exception of the central lesion, the adjacent compartments present some similarity between them but also leading to a certain degree of heterogeneity. We also then generated a heatmap comparing core and internal wall, as requested by the reviewer, and by stratifying against the clinical data. Results showed that there was no clustering of the clinical surrogates with neither the central nor the internal granuloma compartments (Supplementary Figure 3).

Minor comments:

Line 156 - remove diverse, as diverse and diversity are both mentioned in the sentence, doubling up

Answer: As per required by the reviewer, we have removed “diverse”, now in line 185.

Line 179 - significantly should be significant

Answer: As pointed out by the reviewer, we have changed significantly for significant, now in line 205.

It is unclear what the module colours mentioned in the figures add, please explain or remove.

Answer: In WGCNA, module colours represent distinct clusters of genes that are grouped based on similarity in their expression profiles. These clusters are automatically assigned different colours (such as blue, green, yellow, etc.) to visually distinguish them. Each module colour indicates a unique group of genes that may share biological functions, pathways, or regulatory mechanisms. These colours are arbitrary labels without any intrinsic biological meaning used primarily to make it easier to discuss and visualize different gene modules.

Since the information on module colours and their annotations is already provided in Supplementary DataSet 2, we have removed the module colours from the figures. Figures 3c and 4a have been modified accordingly and an explanatory sentence added in Methods clarifying the module colours meaning (lines 696-697).

ANSWERS TO REVIEWER #3

Reviewer #3 Feedback

Dear reviewer, please find here below our answers to your queries and suggestions in red, and changes to the figures highlighted in yellow.

Many thanks in advance,

Yours sincerely,

The authors

This study explores the transcriptional profile of diverse lesions obtained from n=13 patients with pulmonary TB. Bulk RNA-seq of multiple tissue biopsies (n=44) obtained from different sites of TB lesions was used along with weighted correlation network analysis to map diverse microenvironments in the Mtb-infected lungs. The transcriptional profiles were associated to disease severity and clinical outcome. The main findings presented were that more severe lesions displayed enhanced inflammation characterised by IFN-signalling and neutrophil degranulation. Although I sincerely understand and appreciate the efforts invested into this precious tissue material including the impressive RNA analyses, the significant advances obtained from this study are not clear. Enhanced inflammation in severe TB disease along the IFN/neutrophil axis is not new. Instead, the manuscript is entirely void of data describing T cell responses in these different granuloma sites. A significantly more detailed description of the outline and analyses in the MM section is required to understand the specific clinical and pathological criteria used in the study. In addition, the manuscript would benefit from adding some type of spatial image analysis (at the protein level) including cells and markers that could support the RNA-seq data, which would bring significant added value to the study.

Specific comments:

1. While the RNA-seq analysis done is extensive it is also the sole experimental pillar in this study. As multiple tissue biopsies were collected from each patient, the authors should present complementary data on cell types (e.g. macrophages, T and B cells, neutrophils, epithelial cells, fibroblasts) as well as protein expression and distribution in the tissue lesions using multiplex imaging or similar technology. This would bring significant added value to the RNA-seq analyses.

Answer: We thank the reviewer for the valuable suggestions, which have helped improve the quality of our work. Even if we have not been able to conduct a multiplex imaging, we have extended our studies and included more data, specifically:

1. To provide complementary data on specific immune responses, we used the LM22 signature matrix to profile the distinct human hematopoietic cell populations (PMID: 25822800). We have expanded the adaptive/humoral and innate/PRR modules enriched for the genes related to the immune populations and, annotated these submodules to reflect their expression in the TB lesion and in each compartment, resulting in

Supplementary Figures 4 and 5a. We found that these submodules were significantly differently enriched when comparing the compartments (Supplementary Figure 5a).

We have further examined the association of these submodules with the severity correlates, identifying 3 submodules enriched in a statistically significant way (Supplementary Figure 5b):

- a. Response to inflammation, derived from the innate/PRR module, linked to neutrophils and granulocytes, with genes related specifically to response to IL-1 and type II IFN and neutrophils chemotaxis.
- b. Innate response regulation, also derived from the innate/PRR module and linked to neutrophil, monocytes and macrophages, with genes related specifically to immune response activation and regulation.
- c. CD4+ T helper lymphocyte response, derived from the Adaptive/humoral module, linked to antigen-presenting cells and CD4 T cells, with genes specifically related to the regulation of T cell activation, lymphocyte differentiation and regulation of the adaptive immune response.

To include these data and results, we have modified the manuscript text in the following sections: Results lines 247-254, 260-262 and 296-306 of the tracked-changes version provided, Discussion lines 439-446 and Methods lines 712-722. We have also updated the Supplementary DataSet 2 and added Supplementary Figures 4 and 5 to include the submodules data.

2. To gain insight into the pathways that differ from patients with different clinical status, we have conducted an analysis of transcriptional factors, obtaining a set of transcription factors that are differentially expressed between fast and slow sputum culture converters and patients with less severe or more severe symptoms. As a result, we have included this information in the manuscript text, specifically in the following sections: in Results section (lines 307-315), in Discussion Section (449-458, 483-493) and in Methods Section (lines 739-746). We have also added a new Supplementary Table 2 and Supplementary Figure 6.

3. We have conducted immunohistochemistry confirmation of RNA sequencing data from three genes (*CXCL9*, *GPB5* and *STAT1*) significantly expressed between granuloma and NL tissue, and that corresponds to the module with the highest enrichment associated with TB severity (SCC and SGRQ symptoms score). As a result, we have modified the manuscript text in the following sections: in the Results section (lines 315-321), in Discussion section (lines 493-496) and in Methods Section (lines 748-769). We have also added Figure 5.

2. Have a control group been included e.g. representing uninfected individuals who undergo thoracic surgery due to another disease indication such as lung cancer? This would add relevance for comparison to the RNA-seq analysis. Also to the demographic data, where for example BMI is relatively high among the TB patients. Why is that? Wasting and weight loss is one of the primary clinical symptoms of chronic TB, while

the 14 patients in this study are normal weight or obese according to Table S1. This should be explained and discussed and preferentially related to a control cohort.

Answer: We appreciate the reviewer's suggestion to include a control group of uninfected individuals undergoing thoracic surgery, such as patients with lung cancer. To identify a suitable control group in this context is challenging, as healthy individuals rarely undergo lung surgery without an underlying medical condition, and we considered patients with lung cancer but discarded this option because lung cancer itself—and the associated treatments—can significantly alter lung tissue and gene expression profiles, potentially confounding our findings. For all the transcriptomic analysis and during the conception of the study we included the non-lesional tissue (NL) as negative control, even if we acknowledge that using uninvolved lung parenchyma from our cohort participants as a control does not eliminate the possibility of immunological influences from the TB lesion environment affecting these uninvolved areas, which could bias our results. The advantage of using samples from the same patient is that it helps control inter-individual variability. However, we acknowledge that the reviewer is right, and for this reason, we have referred to this control tissue as "non-lesional" and have now included a statement highlighting this limitation in the Discussion section (lines 548 - 553).

Nevertheless, following several reviewers' suggestions, we performed RNA sequencing validation at the protein level in tissue samples by immunohistochemistry (see comments above). For this we also included a negative control consisting of patients diagnosed with bullous emphysema complicated by spontaneous pneumothorax, as closest to healthy people as possible. These results are presented in Figure 5, showing a statistically significant increase of CXCL-9, STAT-1 and GBP5 in TB lesion.

Regarding BMI, the reviewer is correct that wasting and weight loss are among the primary clinical symptoms of active TB. However, it is important to note that all these patients were correctly treated and were operated due to persistent lesions on their X-rays despite achieving microorganism clearance from sputum. This means that there were not in the acute phase of the disease (where the consumption is more evident and therefore there is the highest impact on the BMI). In fact, most of TB patients, in general, tend to rapidly increase their BMI with the drug therapy at the beginning of their clinical management, and that this increase might be faster or slower depending on their background, clinical variables and outcomes (Diallo et al, 2020 PMID: 32345228). This is the reason for the patients presenting such a wide range of BMI values. In order to avoid any confusion and clarify this, we have removed "from critically ill patients" from the discussion section (line 388).

Additionally, were blood chemistry analyses performed that could provide additional information on disease severity e.g. ESR/CRP, WBC, CD4/CD8 T cell counts, Hb, albumin?

Answer: We thank the reviewer for suggesting the inclusion of blood chemistry analyses which could provide valuable information on disease severity. From this cohort we do not have data on ESR, WBC counts, Hb or Albumin, but we do have CRP values, which

were low (median value of 3.99 mg/L). We have added this information in the Supplementary Dataset 2.

3. The Introduction seems to be written in a similar way as compared to the Discussion. I lack a clear introduction and background that clearly describes the aims of the study and that provides a solid background to current knowledge about granuloma composition and structure. Three initial references in the Introduction refer to WHO; is this necessary? Currently, the authors put a lot of focus into citing recent high-profile studies in the field, while I miss out on several original references on TB granuloma imaging and computerized quantification of cells and molecules both the RNA and protein level, such as PMID: 10768979, 11918694, 12379712, 14638800, 17664265, 25510482, 19435796, 19435796, 15376257, 35092727, 31015452, 33552077. This is particularly important as many of these references refer to cellular immunity including T cell responses that are largely missing from the current discussion.

Answer: Following the reviewer's suggestion, we have revised and modified the introduction and discussion sections accordingly. We have modified the last paragraph of the introduction to clarify the aims of the study. We also have deleted the 3 references linked to WHO and replaced them with only one (the last WHO Global TB Report). Regarding the papers the reviewer suggested, we have included in the Introduction and Discussion sections the following articles: PMID: 10768979, 11918694, 14638800, 15376257 as well as 36920308 as we believe they are appropriate and provide valuable information. Changes in discussion section are better explained in our answers to the reviewer next queries.

4. As the patients were selected for adjunct surgery based on their chest X-ray findings and response to treatment, have any type of radiological scoring been used to quantify the extent and severity of pulmonary involvement including the extent and character of the inflammatory lesions, consolidations, level of fibrosis and cavity formation (size and numbers, lung lobe and zone)? There are multiple manual scoring systems available in addition to computer-aided detection of TB. Chest X-ray grading can also be done retrospectively on existing X-rays. Currently, the only data available is "multiple lesions in the CXR less or equal to/more than 2", which provides very limited information.

Answer: We agree with the reviewer's point. However, we used the only radiological data available for this cohort, which includes the most critical radiological indicators correlated with severity: the presence and number of lesions. We could not analyse the data by presence/absence of cavities, as all patients in the cohort presented with cavities on their chest X-rays. However, we conducted an analysis based on the number of cavities, as mentioned in the Results and Methods section, categorizing cases with two or more cavities as more severe, not finding any statistical differences. We also had data on the localization of the lesions in the lung lobes, even if we couldn't use it for further analysis as 11/14 were located in the upper lobe. However, following the reviewer's comment, we have now included all this information available in the Supplementary Dataset 2.

5. How come n=7 DS-TB patients were referred to surgery or what were the specific criteria or indication for surgery? With what method was drug susceptibility of the Mtb isolates tested? A Table or description of the drug-resistance present should be added.

Answer: In Georgia and other former Soviet Union countries, MDR and XDR-TB remain significant public health challenges, therefore therapeutic surgery is commonly used, especially in cases with extensive lung damage, large cavities, or significant radiological infiltration. The WHO has emphasized surgery's role in these cases to remove infected lung tissue and reduce bacterial load, thus improving prognosis (WHO 2014 - The role of surgery in the treatment of pulmonary TB and multidrug- and extensively drug-resistant TB).

All patients recruited in this study received therapy according to national guidelines (WHO 2014 - The role of surgery in the treatment of pulmonary TB and multidrug- and extensively drug-resistant TB) and, at the moment of the surgery, the whole cohort had a bacteriological conversion of sputum and was negative by smear microscopy and culture. The participant inclusion criteria were if patients required surgery because of persistent radiological signs of cavitory lesions in the CXR and computed tomography (CT) scan, according to official guidelines' surgery recommendations. Thoracic surgery indication was made by the NCTLD Resistant Tuberculosis Treatment Committee, composed of two surgeons and 18 pulmonary TB specialists.

At diagnosis of the current TB episode, each person started standard treatment regimen for their pulmonary TB according to Georgia national guidelines for DS-TB, MDR-TB, and XDR-TB strains (WHO 2017 - Guidelines for treatment of drug-susceptible tuberculosis and patient care; WHO, 2019 - Consolidated Guidelines on drug-resistant tuberculosis treatment). During the antibiotherapy treatment course, participants exhibited consecutive negative sputum cultures, as well as negative AFB smear, demonstrating bacilli clearance. However, they still exhibited lesions on their X-rays, regardless of drug sensitivity.

Following the reviewer's comment and to increase clarity, we have revised the section "Study design and patient cohort" in Methods accordingly (lines 582-597).

The drug susceptibility of the Mtb isolates was tested with molecular drug sensitivity testing (Line-Probe assays) or classical phenotypic drug susceptibility testing. The only data available in terms of drug sensitivity was if the TB case was sensitive, multidrug resistant (MDR) or extremely resistant to drugs. Following the reviewer's suggestion, we have added the information on the drug sensitivity in the Methods section and in the Supplementary Dataset 2 ("TB drug sensitivity" column).

6. In the MM section it is described that the St. George's Respiratory Questionnaire (SGRQ) was used to assess the presence of TB symptoms. How come the authors used this score that does not seem to be specifically designed to measure TB symptoms, when there are validated composite TB scores available (e.g. PMID: 17852907, 23113626, 33757875, 24857613)? The variables included in the SGRQ score should be listed and

explained in the manuscript. Currently, there is no information as to what components (relevant to TB disease) this score contains or what a threshold of 20 means (as the score ranges from 0 to 100, correct?).

Answer: We appreciate the reviewer's concerns regarding the use of the SGRQ. Validated composite TB scores, such as the one developed by Wejse and collaborators, would indeed be valuable for assessing disease severity in our patient cohort. We have extensively worked with these scores in other clinical trials and studies, and they are useful. However, due to the lack of available data for this cohort (doi: 10.17632/knhvdbjv3r.1), we were unable to calculate the TB scores retrospectively. To address this limitation, we have used the symptoms component of SGRQ, which was available and assesses the frequency and severity of respiratory symptoms reported by the patients enrolled in the study.

The SGRQ is a health-related quality-of-life tool that has been validated for several types of lung disease, including TB and applied to measure the impact on lung function during TB (Pasipanodya et al 2007, PMID: 17890471; Brown et al, 2015, PMID: 25809759; Gupte et al, 2019 PMID: 31064624; Benito et al 2020, PMID: 32904577; Stringer et al, 2021; PMID: 34489263; Romero et al, 2024 PMID: 38349459). It assesses respiratory health across three main components: symptoms (frequency and severity of respiratory symptoms), activity (limitations on physical activities due to breathlessness), and impact (effects of respiratory issues on overall well-being). Each component is scored, and a total score reflects the overall impact on quality of life. In this study, we used the symptom component of the SGRQ, which assesses the impact of respiratory symptoms as well as their frequency and severity as perceived by patients. To stratify patients into "less severe" and "severe" groups, we set a threshold based on the median symptom score across all participants, which was 20, as mentioned in the lines 725-732 of the Methods section. To improve clarity, we have added an explanation of the variables included in the SGRQ score to the manuscript (Methods section, lines 604-610).

It is also stated that: "TB individuals were classified taking into consideration clinical surrogates of disease severity." Meaning what? SCC and SGRQ symptom score? Was there a difference in the SGRQ score comparing DS- to DR-TB? Was there a correlation between time to sputum culture conversion and SGRQ?

Answer: We used several surrogates of disease severity in our study, related to worse TB outcomes and based on the information available: the SCC, the symptoms component of the SGRQ, the infecting-strain drug sensitivity (DS or MDR/XDR), if those TB cases were relapsed or not, the number of lesions (Methods section, lines 725-732). However, the results showed only significance for SCC and SGRQ (Results section, lines 284-286 and Figure 4).

Regarding the question of the reviewer, and although results were not statistically significant, it seems that individuals with MDR/XDR-TB had a higher SGRQ symptom score compared to cases of drug-sensitive tuberculosis (DS-TB) even if no correlation was found with the time to sputum conversion (see figures below). The following figures were not added to the manuscript.

7. It is unclear for the MM section exactly how the different lesion areas were selected and scored. Only by macroscopic examination by a pathologist? Was HE-stains made to support macroscopic findings? Pls, define “granuloma sample”. Currently, only the size of the lung biopsies is given, but how can one single granuloma be detected and dissected from each piece of tissue as confluent granulomas of different sizes are very common?

Answer: During surgery, TB granuloma lesions were removed (median of 3.2 cm in diameter), and cut to obtain: 1) one piece containing all compartments and non-lesional tissue for pathology studies; and 2) 48 biopsy fragments ($\sim 0.5 \text{ cm}^3$) of tissue samples in RNAlater solution at 4°C overnight, before storage at -80°C for further RNA-Seq analysis. The selection and scoring of the different lesion areas were performed through macroscopic examination, with surgeons and pathologists determining the approach to compartmentalization based on their clinical and pathological expertise. For better clarity purposes, we have better explained this in the Methods section (lines 613-623). A big sample from the removed lesion, not per separate compartment, was then collected for histopathological evaluation, limiting direct comparison with the RNA-seq samples.

Following the reviewer’s question on the term “granuloma sample”, we do agree that it is a term that might not be totally accurate. To solve this, we have substitute it everywhere in the manuscript and figures for TB lesion.

In Table S1. “Fresh vs Not fresh necrosis” does not make sense to me. It is confusing to place parts of the characteristics for the TB lesions in Table S1, and another part of characteristics of TB lesions according to treatment response as Table S2. An explanation to the “category” Cavitation and Tuberculoma is also required.

Answer: We thank the reviewer for the comment, and indeed we do agree that the data we provided on the TB lesions could be confusing. To clarify this but also to answer another of the queries on the pathology details of the TB lesions resected, we have removed the original Supplementary Table 2 and included a complete table on TB lesions and all associated pathology data we have in Supplementary Dataset 2, and the correspondent text in the Results and Methods sections.

8. In Fig 2a, the TB granuloma is depicted in the lower part of the lung, while lesion is normally found in the apical parts of the lung. Please, also specify which lung segments/lobes the biopsy material was obtained from. In Fig 2a, it would also be helpful to list, for each segment, C, I, E or NL, the criteria/features used for dissecting each tissue section into a separate vial for the RNA analyses. In addition, the colour scheme becomes a bit too much with all the colours depicted, e.g. “Granuloma” could perhaps be described in plain black or grey text (and colour in Fig 2b).

Answer: The reviewer was right about the location of the granuloma lesion. Consequently, we have amended Fig. 2a which now depicts the lesion in the apical part of the lung. We also have added the data on from which lung lobes the biopsy material was obtained in Patients’ data tab of Supplementary Dataset 2.

Details on the criteria used to separate each tissue section for RNA analyses are now better explained in the Methods section (lines 613-623).

Following the reviewer’s suggestion, have also adjusted the TB lesion colour to a grey scale, in both figures 2a and b.

Pls, list the individual RIN values for each tissue specimen in Supplementary Data set 2.

Answer: We have included the RIN values for all the samples in the Supplementary DataSet 2 (Patient Data tab).

I cannot find any information on gene expression modules including T cell responses in any of the tissue specimen analysed, why is that?

Answer: We thank the reviewer for this observation. To fulfil this gap, we have included data regarding the gene expression including cell populations responses. The LM22 immune population transcriptomic signature (PMID: 25822800) was used to profile the distinct human hematopoietic cell populations and know the location of genes associated to them within the WGCNA modules. Consequently, we expanded the two largest immune-associated modules with the highest number of genes associated with distinct human hematopoietic cell populations, the humoral and innate/PRR modules. Within these modules, we could identify sub-modules related to distinct immune processes and responses. This in-depth analysis and findings are now included as Supplementary Figures 4 and 5. We also have re-annotated the humoral module to “Adaptive/Humoral” to include the T cell responses and to be more accurate. The text has also been amended to include all these new data in the Results (lines 247-254, 260-262 and 296-306), Discussion (lines 439-446) and Methods sections (lines 712-722).

9. The Discussion could be rewritten to include a more integrated mechanistic explanation on the humoral response, DNA binding and myeloid activation modules together with the IFN and neutrophil modules detected in the lesions. Nothing is mentioned about T cell responses that are known to be crucial for Mtb control, can the authors please comment on that? Overall, the results are presented broadly now, and the manuscript could benefit

from some focus that ties all the loose ends together, perhaps also using a graphical illustration of the main conclusions. In several places in the Discussion, the authors indicate that responses observed in peripheral blood reflect the microenvironment at the site of infection. This is debatable, and I would strongly recommend lifting the advantages of studying responses at the granuloma sites.

Answer: Following the reviewer's suggestions, we have worked on the discussion and improved it substantially.

As requested, we have included the immune processes and responses analysis in the discussion, and we also have deleted the sentence stating that blood reflects the microenvironment at the site of infection, as per the reviewer's suggestion.

Finally, we have created a graphical illustration of the main conclusions of the paper, now included as Figure 6, which indeed we do believe helps as a take-home-message.

10. Minor comment: The manuscript needs language editing. For example, second sentence in the abstract "... 13 patients undergoing therapeutic surgery using RNA-Sequencing." is not correct. Overall, there are numerous text parts and sentences throughout that appear misplaced and disturb the reading flow of the paper.

Answer: Following the reviewer's suggestions, we have carefully considered all suggestions and have made extensive revisions and corrections to enhance clarity and improve the overall quality of the manuscript.

ANSWERS TO REVIEWERS

ANSWERS TO REVIEWER #1

The authors have not completely addressed my concern regarding the usefulness of comparing the RNA-seq data to sputum conversion and disease severity given that all TB individuals recruited to the study underwent therapeutic surgery and must have presented with severe immunopathological disease. How will the findings inform the treatment plan? A modified discussion to address this would be helpful.

We understand the reviewer's concern regarding the relationship between clinical signs and treatment adjustment given the characteristics of this cohort of patients. With our study, we were able to associate aggravated conditions within the TB lesion to clinical surrogates of disease severity. The fact that worsened clinical signs are linked to such conditions suggests that an opportunity of adjusting treatment is available whenever these signs are present. To clarify this, we have modified the Discussion lines 451-461 of the manuscript version with tracked changes, which now reads as follows:

“Therefore, achieving SCC after two months of starting treatment has been associated with TB cavity persistence⁶⁰ and poor prognosis^{61,62}, and has been proposed and used as a surrogate marker for TB outcomes. Now, thanks to our study, we demonstrate that a sustained pro-inflammatory profile at the lesion site is linked to delayed culture conversion (persisting beyond 8 weeks post-treatment initiation) focusing on a cohort of patients with severe immunopathological disease who underwent therapeutic surgery. This finding implies that delayed culture conversion may serve as a proxy for heightened lesion inflammation and, by extension, worse clinical outcomes. It may thus facilitate the identification of patients who could benefit from enhanced therapeutic strategies, including the incorporation of anti-inflammatory host-directed therapies into standard treatment regimens.”

ANSWERS TO REVIEWER #3

1. I believe the authors have addressed most of my concerns and significantly improved the manuscript, particularly by including several clarifications in the Materials and Methods section, performing immunohistochemistry (IHC) on selected proteins, and utilizing the LM22 immune population transcriptomic signature to profile human hematopoietic cell populations in the lesions. However, it has also become clear that the patients included in this study were not subjected to surgery due to treatment failure but were considered microbiologically cured. This is important information regarding their clinical TB disease and should be further emphasized in the manuscript.

Even if the reviewer is right and the patients included in this study were not subjected to surgery strictly due to treatment failure and they all were considered microbiologically cured, they were subjected to surgery due to persistent signs of cavitory lesions. Even if this was already mentioned in Methods section lines 523-527 of the manuscript version with tracked changes, we do agree with the reviewer that this information regarding their clinical TB disease is important and we have emphasized it in the manuscript, specifically in the Discussion section lines 466-471, as follows:

“These TB individuals presented advanced TB disease rendering them candidates for lung resection surgery despite being microbiologically cured. Notably, the surgery was performed not because of treatment failure but to address persistent cavitory lesions, which may limit the generalizability of the findings to the entire spectrum of TB disease.”

2. Even though the patient selection, clinical data collection and biopsy sampling are clearer now, I suggest adding a sentence in the first paragraph of the Results that a normal to high BMI, low CRP and relatively low SGRQ score, were consistent with microbiological cure and that these patients were not considered critically ill. I believe this is important information as the readers may otherwise be misguided.

As per the reviewer’s suggestion, we have added the requested information in the first part of Results section lines 125-127:

“Although the patients included in this study exhibited normal to high BMI, low CRP levels, relatively low SGRQ scores and were considered microbiologically cured, they nonetheless required lung resection surgery due to the persistence of TB cavities.”

3. Include data in the MM section from line 723, indicating that the Saint George’s Respiratory Questionnaire (SGRQ) symptom scores range from 0 to 100, with higher scores indicating more respiratory limitations. If there is no available information on this score, what is assessed and how it is ranged, it is not possible to critically understand an SGRQ score below or above 20. I also recommend listing the symptoms components of the SGRQ for each patient that was used in this study as a Supplementary data set (perhaps as part of S. data set 2).

As per the reviewer's suggestion, we have added the requested information in Methods section lines 656-661. We have also included the list of the symptom components of the SGRQ for each patient that was used in this study as a new tab in Supplementary Data Set 2.

“The SGRQ symptoms scores comes from an eight-item questionnaire with a weighted score ranging from 0 to 100, with higher scores indicating higher effects, frequency and severity of respiratory symptoms (Supplementary Data Set 2). To divide the patients for the analysis hereby presented we used a cut-off defined by the median SGRQ symptoms value, >20 being considered more severe.”

4. It is unclear why the authors omitted lines 270-276? Without this information in the Results, it will be difficult for the readers to follow.

We had included this information in Methods section. However, following the reviewer's comment we have included it back to Results section lines 236-237 and explained it better in Methods section as per the previous comment.

5. Thank you for adding the RIN values in S. data set 2. I do understand that a lower RIN is to be expected in this type of tissue material, but a RIN value of 4-7.5 is generally considered moderate and may be acceptable for some RNA analyses, but it is not ideal for highly sensitive applications like RNA-seq where a RIN < 8 is typically recommended. Please, briefly comment on this limitation in the manuscript and add information on the RIN values in the MM section of the main manuscript to clarify that the RIN in the different biopsy samples ranged from 4-7.4.

As per the reviewer's suggestion, we have included this information as a limitation in Discussion section lines 485-487 and in Methods section lines 563-566.

“Another limitation is that given the source and status of the human TB lesion samples, it was required to lower the RINe cut-off to four, acknowledging this a potential bias in RNA-seq experiments.”

6. It could assist the readers if the authors could highlight the regions depicted in the images included in the new Fig. 5, eg. necrotic core (NC), lymphocyte rim, alveolar space etc.

Following the reviewer's suggestion, we have highlighted the regions depicted in the new Figure 5, shown below.

Figure 5. Immunohistochemistry staining of representative genes associated with TB severity reveals higher protein expression in TB compared to non-TB controls. Panel a shows representative immunohistochemistry staining for CXCL9, GBP5, and STAT1 from the TB lesion of a representative patient (TB-05) compared to a patient presenting bullous emphysema (TB-42), as non-TB control. The top row corresponds to whole sections of the TB lesion (at the left of the images) and of non-lesional tissue (at the right of the images). Scale bars correspond to 1000 μm . NC = necrotic core, M = macrophage region, F = fibrotic region, L = lymphocyte-enriched region, AS = alveolar space. **Panels b, c, and d** show the quantification of CXCL9, GBP5, and STAT1 protein levels respectively in all TB patient granuloma sections (n=14) compared to the non-TB control tissue sections (n=3). Data on the percentage of stained area are represented as median with an interquartile range. Statistical analysis was performed by applying the Wilcoxon-rank sum test. Statistical differences refer to a p -value < 0.05.

- I think that the addition of the data on specific immune responses using the LM22 signature matrix is interesting and beneficial to the data processing and presentation. Would it also be possible to use the LM22 transcriptomic matrix to plot the relative percentage of each immune cell subset plotted in Supplementary Figure 3, comparing the relative immune cell distribution in the TB lesion vs non-lesional tissue and an also external wall to central wall? These data could be presented eg. in an overlay bar chart

b
Supplementary Figure 3. Immune populations associated with modular transcriptomic signature in the TB lesion and their distribution among TB lesion compartments. Panel a depicts the distribution of the LM22 signature and the corresponding immune populations among the TB lesion modules. Color intensity represents the number of LM22 genes associated with the corresponding immune population present in each TB lesion module. EMT = Epithelial to Mesenchymal Transition; PRR = Pattern Recognition Receptors. **Panel b** shows the proportional distribution of LM22 immune populations across non-lesional tissue (NL) and TB lesion external (E), internal (I), and central (C) compartments, as obtained from CIBERSORT deconvolution.

8. In S. Figure 4, I find it interesting that the humoral module appears mostly upregulated, even compared to IFN-signaling. This should be highlighted in the Results of the manuscript (starting from line 245).

Could the authors also plot Turquoise 4 and 7 comparing the symptoms score (similar to S. Fig. 4b) as well as comparing fast vs slow SCC? Could this mean that stronger B cell activation including an antibody response in the lesions is associated with a dysregulated immune response locally in the lesions? Perhaps there is a lot of

extracellular bacilli in the lesions of these “microbiologically cured” patients that drive inflammation and particularly B cell activation?

Indeed, it is very interesting that the humoral module appears mostly upregulated, and we have highlighted this in the Results section lines 217-218. Regarding the reviewer’s request, we had already tested all the submodules (including Turquoise submodules 4 and 7) for differences when comparing SCC and SGRQ. The only ones that showed a statistically significant difference were those presented in Supplementary Figure 4b. We have modified Results section lines 255-257 to clarify this matter.

“Further examination of the associations between submodules and severity correlates revealed that only three submodules exhibited statistically significant differences among clinical surrogates”

The stronger B cell activation was not found to be associated with the SCC or SGRQ status. Regarding the inquiry about the presence of extracellular bacilli, unfortunately we are unable to compare the results of the Ziehl-Neelsen staining of the compartment samples with B cell activation as separating by compartment reduces the sample size to a degree where no appropriate conclusions can be drawn. Although we cannot draw any strong conclusions on the role of this stronger B cell activation with the data we have in this study, as per a previous comment we have included the following in the Discussion section lines 362-369:

“Interestingly, we found that the adaptive/humoral module was increased in whole TB lesion samples, corroborating the expression of immunoglobulin heavy and light chains transcripts in both central and internal compartments, as well as the higher proportion of effector B cells across the TB lesion. The enrichment of the adaptive/humoral module, along with increased lymphocytes—particularly effector B cells—in TB lesions, suggests an elevated antibody response. This is consistent with recent findings by Krause et al., who reported abundant B cells and high levels of Mtb-reactive antibodies in these lesions.”

9. Lines 355-358 in the Discussion: How can these results be used to refine anti-TB treatment? Using HDT or something else? Please, briefly develop/clarify.

Yes indeed, HDT mainly with anti-inflammatory effect. We have expanded this part of the Discussion section lines 451-461 to clarify this point and include how the treatment could be adjusted – with host-directed therapies for sure. The paragraph added is:

“Therefore, achieving SCC after two months of starting treatment has been associated with TB cavity persistence⁶⁰ and poor prognosis^{61,62}, and has been proposed and used as a surrogate marker for TB outcomes. Now, thanks to our study, we demonstrate that a sustained pro-inflammatory profile at the lesion site is linked to delayed culture conversion (persisting beyond 8 weeks post-treatment initiation) focusing on a cohort of patients with severe immunopathological disease who underwent therapeutic surgery. This finding implies that delayed culture conversion may serve as a proxy for heightened lesion inflammation and, by extension, worse clinical outcomes. It may thus facilitate the identification of patients who could benefit from enhanced therapeutic strategies, including the incorporation of anti-inflammatory host-directed therapies into standard treatment regimens.”

10. Line 444 in the Discussion: I realize the module data shown in S. Figure 4a, supports the assumption that T cells are enriched in the central and internal compartment of the TB lesions. However, most literature on human TB granulomas, including studies from the groups of JL Flynn, G Kaplan, and T Ulrichs etc, features lymphocytes in the peripheral rim of the granuloma. These lymphocytes surround Mtb-infected macrophages, possibly preventing the effective killing of infected cells. Can the authors comments on this and adjust their Discussion accordingly?

We acknowledge the reviewer's concern regarding the apparent location of T cells. Although Supplementary Figure 4a shows enriched T cell modules in the central and internal compartments, this reflects a relative enrichment of associated modules rather than an absolute increase in cell numbers. With the addition of Supplementary Figure 3b, the relative proportions of each cell population across compartments are now clearer and align with previous studies by Kaplan et al., Sawyer et al., and Ulrichs et al. Specifically, the external compartment exhibits a higher abundance of lymphocyte populations, whereas macrophage populations are more prevalent in the central and internal compartments. To address this issue, we have provided clarification in the Discussion section lines 371-386.

“Furthermore, we showed that these responses and the proportions of their related immune cells are distributed in a gradual manner across various compartments, which appears to play a central role in the modular signature of TB lesions here unveiled. Overall, the enrichment of immune-related modules mostly in the central and internal compartments and the observed cell distributions suggest that, although the central compartment is predominantly necrotic, it may still harbour a lymphocyte component from adjacent tissue. However, due to the lower overall cellularity in this central compartment, the relative abundance of these populations appears higher, reflecting a greater proportion of T cell-associated signals rather than an increased absolute number of T cells. The relative proportions of each cell population across compartments align with previous studies^{16,17,45}, as the outer portion of the lesion exhibits a higher abundance of lymphocyte populations, whereas macrophage populations are more prevalent in the inner compartments.”

11. Finally, language, spelling and abbreviations needs to be checked (some examples provided below):

- Line 300: strange order of words

We have changed the sentence, which now reads at lines 262-263:

“We also observed an enrichment in the CD4+ T helper lymphocyte response submodule.”

- Line 742: sentence is odd

We have changed the sentence, which now reads at lines 673-674:

“This yielded a list of 92 transcription factors which was then tested using the Wilcoxon-rank sum test.”

- Line 720: incorrect?

Yes indeed, it was incorrect and p-value had to be deleted. Now the sentence reads at lines 647-648:

“employing the three compartments compared to non-lesional tissue as before and considering as significant an $FDR < 0.1$.”

- S. Fig. 4b: upper left graphs, SQRQ instead of SGRQ

We have corrected this in the Supplementary Fig 4b.

- Reply to question 1: incorrect Supplementary Figures 4 and 5a-b are referred to, perhaps authors mean S. Figure 3 and 4?

Yes indeed, we meant Supplementary Figure 3 and 4 and we apologize for this confusion.

Manuscript ID: NCOMMS-24-28326C

Title: Unravelling the transcriptome of the human tuberculosis lesion and its clinical implications

Version 3 – Third round of reviews

REVIEWERS' COMMENTS

Reviewer #1 (Remarks to the Author):

My concerns have been addressed.

Reviewer #3 (Remarks to the Author):

I thank the authors for the revised version of the manuscript and can confirm that all my remaining concerns have been appropriately addressed.